# A Learning Law: Generalization via Geometric Complexity and Algebraic Capacity

## Abstract

Modern machine learning systems achieve strong performance but remain data-hungry and opaque. We propose the *Learning Law*, which asserts that effective learning follows the order *form → law → data → understanding*. We formalize this by separating geometry discovery, law formation, and data calibration. The first stage learns a latent manifold with controlled intrinsic dimension and smoothness. The second restricts predictors to an algebraically constrained law space on this geometry. The third calibrates these laws on finite labeled data. We derive a Geometry–Algebra Generalization Bound showing that population risk depends on geometric complexity $\mathcal{C}(\phi)$ and algebraic capacity $\mathcal{A}(g)$, rather than raw parameter count, yielding intrinsic sample-efficiency advantages for geometry-first learning. A two-stage V-GIB implementation confirms these predictions on CIFAR-10 and a tabular classification task. Geometry-first pretraining lowers intrinsic dimension, improves low-label test accuracy, and outperforms data-first baselines once training stabilizes, with ablations isolating the roles of smoothness and intrinsic-dimension control.

## 1 Introduction: A Paradox in How Machines Learn

Over the past decade, deep neural networks and transformers have reached or surpassed human-level performance on a wide range of perceptual and cognitive benchmarks, from image recognition and speech to machine translation and large-scale language modelling Goodfellow et al. (2016); Kaplan et al. (2020). Yet this success comes with a striking asymmetry. Precisely, state-of-the-art models typically require millions to billions of labeled examples and massive compute budgets, and they often remain brittle under distribution shift and difficult to interpret in mechanistic terms Geirhos et al. (2020); Doshi-Velez & Kim (2017). Humans and other animals, by contrast, acquire robust concepts from very few examples. Infants rapidly form new categories, infer causal structure, and generalize in ways that current models still struggle to match, even when trained on orders of magnitude more data Lake et al. (2015); Gopnik et al. (1999). Few-shot and meta-learning methods attempt to narrow this gap Vinyals et al. (2016); Finn et al. (2017); Snell et al. (2017), but they largely operate within the same basic paradigm, i.e., learning is defined as fitting flexible parametric function classes to large sample sets, with generalization controlled indirectly through regularization and optimization.

This leads to a simple but fundamental question;

> **Question.** If humans learn quickly, robustly, and interpretably from limited data, what structural ingredient is missing from our current machine learning formulation?

In this work we argue that the missing ingredient is not another architectural variant or training heuristic, but *the order in which form, laws, and data interact during learning.*

## 1.1 A different starting point: form before data

There is now extensive evidence from developmental psychology and cognitive science that natural learning systems do not begin from an unstructured tabula rasa. Rather, coarse geometric and structural priors appear early, and detailed content is layered on top of this scaffold.

Infants, for example, segment the visual world into approximately rigid, bounded objects before they know specific categories or names Spelke (1990); Johnson & Aslin (1995). They are sensitive to shape and continuity, preferring coherent, smoothly moving entities over fragmentary motion. In language, children acquire aspects of grammar and word order regularities long before they possess large vocabularies, relying on powerful statistical learning abilities over structured input streams Saffran et al. (1996); Gopnik et al. (1999). These observations suggest a qualitative order: *first recover (or assume) structure, then refine that structure with data.*

A parallel story appears in successful machine learning practice, even if it is rarely framed in these terms. Many of the most effective architectures hard-wire geometric or algebraic structure. Convolutional networks build in translation symmetry, graph neural networks encode permutation invariance, and geometric deep learning explicitly exploits manifold or group structure Bronstein et al. (2017). Manifold learning and representation learning more broadly assume that high-dimensional observations concentrate near low-dimensional, geometrically regular sets Tenenbaum et al. (2000); Belkin & Niyogi (2006). In each case, a notion of *form*—a shape or symmetry of the hypothesis space—is fixed first, and only then are model parameters tuned using data.

Motivated by these converging lines of evidence, we propose to take this ordering seriously and elevate it from practice and intuition to an explicit learning principle.

## 1.2 The Learning Law (informal statement)

We introduce the following informal law to capture the qualitative structure of effective learning systems:

**The Learning Law (informal).** *Effective learning proceeds in the order*

$$\text{form} \;\rightarrow\; \text{law} \;\rightarrow\; \text{data} \;\rightarrow\; \text{understanding}.$$

In this;

1. *Form* denotes the geometry or structure of the space in which learning takes place, that is, manifolds, graphs, symmetries, partitions, or other constraints that shape possible representations, independently of any particular dataset.

2. the *Law* denotes the family of relations that are admissible on this form, e.g., dynamical rules, conservation laws, invariances, causal constraints, or algebraic relations that govern how quantities may co-vary.

3. *Data* denotes finite observations that select and calibrate a specific instance of the law within the allowed family, by adjusting parameters rather than redefining the underlying form.

4. *Understanding* denotes the resulting compressed, structured representation that supports robust generalization, prediction, and intervention; in our setting, this will be made precise through information–geometry trade-offs.

Current mainstream machine learning pipelines often invert this order. They begin with large datasets, search over highly expressive function classes with relatively weak geometric priors, and only then attempt to extract structure post hoc through interpretability tools or fine-tuning. The Learning Law suggests the opposite design philosophy, that is, start by specifying or discovering appropriate *form*, constrain admissible *laws* on that form, and use *data* primarily to localize within this structured space. In the remainder of this work we make this idea precise, connecting it to geometric representation learning and information-theoretic objectives, and show how it leads naturally to more data-efficient and interpretable learning rules.

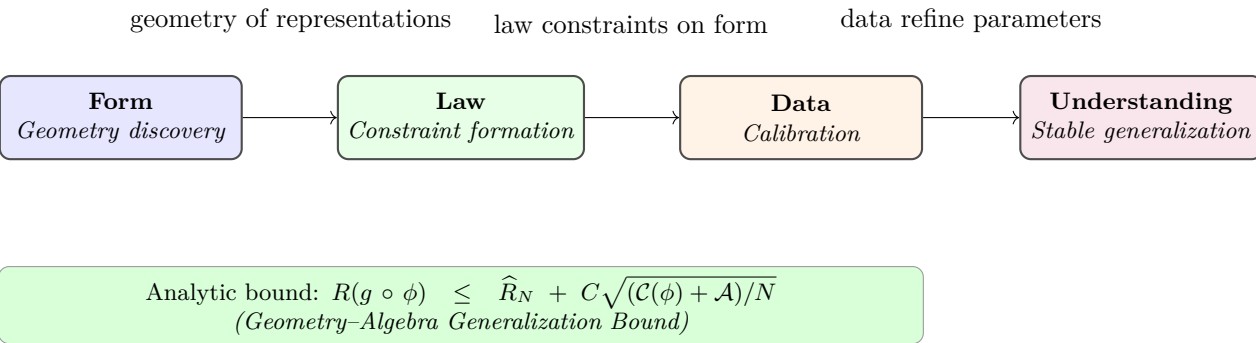

Figure 1: **The Learning Law:** vertical flow of geometry-first learning for two-column layouts. Learning proceeds from *form* (geometry) to *law*, to *data calibration*, culminating in *understanding*. Each stage constrains the next, reducing geometric complexity $\mathcal{C}(\phi)$ and algebraic capacity $\mathcal{A}$.

## 2 Human vs Machine Learning

Human and machine learning both transform experience into structure and prediction, but they appear to do so in systematically different ways. Converging evidence from neuroscience and cognitive science suggests that biological systems rely on strong structural priors and internal geometries that emerge early, such as object structure, spatial maps, and approximate laws of motion, well before rich, labeled experience is available Marr (1982); Spelke (2013); Carey (2009); Dehaene (2011); Landau et al. (1988); O'Keefe & Dostrovsky (1971); Hafting et al. (2005); Moser et al. (2008). Contemporary machine learning, by contrast, typically begins with large datasets and highly flexible function classes, letting representation geometry and implicit *laws* arise as byproducts of empirical risk minimization Goodfellow et al. (2016); Bengio et al. (2013); Kaplan et al. (2020); Geirhos et al. (2020). In this section we juxtapose these regimes. Section 2.1 reviews evidence that brains tend to follow a form → law → data progression, while Section 2.2 shows how dominant machine-learning practice largely inverts this order.

**Remark 2.1** (Significance)**.** *This framework unifies insights from cognitive science and geometry-aware learning into a single analytic principle, that is, that robust intelligence emerges when learning proceeds from structured form to lawful constraint before data calibration. It reframes generalization as a property of geometric and algebraic structure, offering a mathematically verifiable route toward data-efficient, interpretable, and auditable machine learning.*

### 2.1 How Brains Learn: Evidence for Form → Law → Data

In Section 1 we argued, at a conceptual level, that effective learning in natural systems appears to follow the order

$$\text{form} \; \rightarrow \; \text{law} \; \rightarrow \; \text{data} \; \rightarrow \; \text{understanding}.$$

We now show that this ordering is not merely philosophical. Across perception, development, and memory, contemporary neuroscience and cognitive science support the view that the brain first constructs geometric scaffolds and structural regularities, and only then uses sensory experience to refine parameters within this structured space.

#### 2.1.1 Perception: edges, shapes and surfaces before objects

Classic work in visual neuroscience showed that early visual cortex encodes simple geometric primitives. Neurons in primary visual cortex (V1) are tuned to oriented edges, bars and spatial frequency, responding selectively to local contrast and orientation rather than to whole objects Hubel & Wiesel (1962). As one ascends the ventral visual hierarchy, receptive fields become progressively larger and more complex, integrating these primitives into contours, textures and shape fragments, and only later into object- and category-selective responses in inferotemporal cortex Felleman & Van Essen (1991); DiCarlo et al. (2012).

This supports a form-first view. The visual system does not start from arbitrary pixel-level statistics but from an architecture that explicitly represents geometric structure—edges, orientations, surfaces and motion—on which higher-level object representations are built. In the language of The Learning Law, early vision establishes a geometric *form* for the visual world; subsequent layers learn *laws* and categories constrained by this form.

### 2.1.2 Development: infants learn structure before content

Developmental studies reveal that infants acquire coarse structural knowledge about the world before they master rich semantic content. By a few months of age, infants exhibit expectations about object permanence, solidity and continuity. They are surprised when objects appear to pass through barriers or vanish without occlusion, even though they cannot yet label or name those objects Baillargeon (1987); Carey (2009). Similarly, infants show sensitivity to approximate numerosity and spatial layout long before formal counting or schooling Dehaene (2011). In language, children display a robust *shape bias*. New words are preferentially generalized on the basis of object shape rather than colour or texture, indicating that geometric form is a privileged cue in early word learning Landau et al. (1988). They also acquire regularities of syntax and word order with relatively small vocabularies, suggesting that structural templates for grammar emerge before the lexicon is dense. Taken together, these findings align with the idea that the learner's first job is to carve the world into stable forms and relational patterns, and only then to attach detailed labels and content.

### 2.1.3 Predictive coding, maps and internal geometry

Theories of predictive coding and generative modelling in the brain further support a form–law–data ordering. In predictive coding accounts, cortical circuits are assumed to maintain an internal generative model of the world and to use sensory inputs primarily as prediction errors that update this model Rao & Ballard (1999); Friston (2010). The internal model itself—its latent variables, their topology and lawful dynamics—functions as a structured hypothesis space, while data act mainly to adjust the parameters within that space. Spatial memory provides a particularly concrete example of internal geometry. Place cells and grid cells in the hippocampal–entorhinal system encode positions and metric relationships within a low-dimensional cognitive map of space O'Keefe & Dostrovsky (1971); Hafting et al. (2005). These maps organize experience. Trajectories, landmarks and episodes are encoded relative to an underlying geometric coordinate system rather than as independent snapshots Moser et al. (2008). Here, the *form* is the internal metric space, the *laws* are the transition dynamics and constraints on movement, and sensory inputs merely refine the current location and map parameters.

Across perception, development and memory, a coherent picture emerges. Early visual areas emphasize edges and surfaces before categories; infants demonstrate object and number structure before rich semantics; and predictive coding and spatial maps suggest that the brain maintains structured internal geometries whose parameters are updated by experience. In each case, learning proceeds from *form* (geometric and structural priors), through *laws* (constraints and dynamics on that form), to *data*-driven refinement. This empirical pattern is precisely the ordering articulated by The Learning Law, and it provides a biological and cognitive motivation for rethinking machine learning objectives so that geometry and structure play a primary, not secondary, role.

### 2.2 How Machines Learn Today: Data → Parameters → Emergent Form

The previous sections suggested that biological learning systems tend to follow an order

$$\text{form} \;\rightarrow\; \text{law} \;\rightarrow\; \text{data} \;\rightarrow\; \text{understanding},$$

which we called the *Learning Law*. Contemporary machine learning, by contrast, largely inverts this order. Models start from data and a loss, adjust parameters by optimization, and only then exhibit internal geometric structure as a byproduct. In this section we make that inversion precise.

### 2.2.1 The standard supervised learning pipeline

In the canonical supervised setting, we observe i.i.d. samples $\{(x_i, y_i)\}_{i=1}^{N}$, from an unknown distribution $P_{X,Y}$, choose a parametric family $\{f_\theta : \theta \in \Theta\}$, and minimize the empirical risk

$$\widehat{R}_N(\theta) \;=\; \frac{1}{N} \sum_{i=1}^{N} \ell\big(f_\theta(x_i), y_i\big), \tag{1}$$

possibly with an explicit regularizer,

$$\theta^\star \;\in\; \arg\min_{\theta \in \Theta} \left\{ \widehat{R}_N(\theta) \;+\; \lambda\, \Omega(\theta) \right\}, \tag{2}$$

where $\ell$ is a task loss and $\Omega$ typically constrains parameter norms or margins Vapnik (1998); Shalev-Shwartz & Ben-David (2014). This empirical risk minimization (ERM) framework underlies most modern deep learning, whether the architecture is a convolutional network, a transformer, or a physics-informed network Goodfellow et al. (2016). The internal representation $z = \phi_\theta(x)$ is defined implicitly by the choice of architecture and the optimized parameter $\theta^\star$, but the objective in equation 1–equation 2 does not make any *explicit* reference to the geometry of the representation space. There is no term that directly controls curvature, intrinsic dimension, or topological structure of $\phi_\theta(X)$. Variants such as self-supervised learning and reinforcement learning preserve the same ordering. The objective is still specified directly in terms of data (or reward sequences), and representation geometry emerges only indirectly through optimization dynamics.

### 2.2.2 Emergent representation geometry and post-hoc analysis

Despite the data-first formulation, deep networks are known to induce rich internal geometries. Activations in intermediate layers define embeddings in which samples cluster by class, exhibit approximate low-dimensional structure, and sometimes linearize semantic factors Bengio et al. (2013). However, these properties are typically studied *after* training, using post-hoc tools such as t-SNE van der Maaten & Hinton (2008), UMAP McInnes et al. (2018), probing classifiers Alain & Bengio (2017); Hewitt & Manning (2019), or representation-similarity measures such as centered kernel alignment (CKA) Kornblith et al. (2019). In other words, geometry is not an explicit design target but an emergent consequence of minimizing equation 1. We first fit the model to data, and only then inspect the learned latent space to understand its structure. Conceptually, this is the reverse of the Learning Law, that is, structure is not a prior scaffold that constrains learning, but a posterior artifact extracted from a trained network.

### 2.2.3 Consequences of data-first learning

The data $\rightarrow$ parameters $\rightarrow$ emergent-form paradigm has several well-documented consequences, such as;

**Data hunger**   Without strong structural priors on representation geometry, models must infer both the task and the underlying structure of $P_{X,Y}$ from finite samples. Empirically, state-of-the-art vision and language systems require extremely large datasets to approach human-level performance, and their behaviour follows scaling laws in which error decreases smoothly only as data and model size grow by orders of magnitude Kaplan et al. (2020); Hoffmann et al. (2022). In the absence of explicit geometric constraints, much of the training signal is spent discovering a useful internal space from scratch.

**Brittle generalization**   When structure is learned implicitly from the training distribution, models can latch onto shortcuts and spurious correlations that fail under distribution shift Geirhos et al. (2020). Small perturbations to inputs can cause large changes in predictions, as seen in adversarial examples Szegedy et al. (2014); Goodfellow et al. (2015), and performance often degrades sharply on carefully constructed test sets drawn from slightly different distributions than the original training data Recht et al. (2019). These phenomena indicate that the learned geometry does not robustly encode the underlying form and laws of the domain; instead, it reflects a finely-tuned fit to the observed data manifold.

**Opacity and unmeasured understanding** Because representation geometry is not part of the objective, it is not governed by an explicit quantity analogous to equation 1. Post-hoc interpretability methods (saliency maps, feature attributions, probing) offer partial insights but do not yield a scalar measure of "how well the model understands" a domain Doshi-Velez & Kim (2017); Rudin (2019). Understanding, in the sense of stable, structured and human-usable internal representations, remains an emergent and largely uncontrolled property. These limitations motivate a complementary, geometry-first perspective. Thus, rather than starting from data and asking what form emerges, we seek learning principles in which *form and law are specified or discovered first*, and data primarily calibrate parameters within that structured space. In later sections, we formalize such a perspective by turning geometry itself into an explicit part of the learning objective.

## 3  The Learning Law: A Principle of Geometry-First Learning

The previous sections contrasted biological learning—which appears to follow *form → law → data*—with standard machine learning, which typically follows *data → parameters → emergent form*. We now make this contrast explicit by formalizing a generic learning pipeline and stating the *Learning Law* as a design principle. Figure 1 illustrates the proposed Learning Law pipeline, framing learning as a progression from geometric form to lawful constraint, to data calibration, and ultimately to stable understanding—a structure that parallels how both human and machine systems acquire and refine knowledge.

### 3.1  Formalizing the learning pipeline

We distinguish five abstract spaces:

- $\mathcal{X}$: raw input space (e.g., pixels, waveforms, tokens);

- $\mathcal{S}$: *form space*, a structured latent space or manifold;

- $\mathcal{L}$: *law space*, encoding constraints, invariants, or causal relations on $\mathcal{S}$;

- $\mathcal{D}$: finite data (observed samples and supervision);

- $\mathcal{U}$: a space of *understanding artefacts*, such as usable predictors, controllers, or generators.

A general learning system is represented by three maps

$$F : \mathcal{X} \to \mathcal{S}, \qquad G : \mathcal{S} \to \mathcal{L}, \qquad H : (\mathcal{S}, \mathcal{L}, \mathcal{D}) \to \mathcal{U}. \tag{3}$$

Informally,

1. *F* extracts a *geometry of form* from raw observations (for example a latent manifold, graph, or feature space).

2. *G* expresses *laws* on that geometry (for example symmetries, conservation relations, or causal mechanisms defined over $\mathcal{S}$).

3. *H* uses finite data $\mathcal{D}$ to *calibrate* free parameters within this structured space, yielding an effective artefact in $\mathcal{U}$.

This factorization is not merely schematic. At the level of hypothesis classes, it corresponds to assuming that predictors factor through $(\mathcal{S}, \mathcal{L})$. Concretely, consider a model class

$$\mathcal{H} = \{x \mapsto h_\theta(x, \mathcal{D}) : \theta \in \Theta\}. \tag{4}$$

We say that $\mathcal{H}$ admits a *form–law factorization* if there exist families $\mathcal{F}, \mathcal{G}, \mathcal{H}_{\text{cal}}$ such that for every $\theta \in \Theta$ there are $F_\theta \in \mathcal{F}, G_\theta \in \mathcal{G}, H_\theta \in \mathcal{H}_{\text{cal}}$ with

$$h_\theta(x, \mathcal{D}) = H_\theta\Big(F_\theta(x), G_\theta(F_\theta(x)), \mathcal{D}\Big), \qquad x \in \mathcal{X}. \tag{5}$$

Any architecture built from an encoder, a structured law module, and an output head (for example encoder–dynamics–readout or encoder–invariant–classifier decompositions) has this form after identifying $\mathcal{S} = F_\theta(\mathcal{X})$ and $\mathcal{L} = G_\theta(\mathcal{S})$. In this sense, equation 3 does not restrict expressivity; it makes explicit the intermediate objects that typical deep models already implement implicitly.

Standard deep learning pipelines implicitly bundle $F$, $G$, and $H$ into a single parametric map $f_\theta : \mathcal{X} \to Y$, where the internal representation and any regularities it encodes are emergent properties of the optimized parameters $\theta$ Goodfellow et al. (2016); Bengio et al. (2013). In contrast, the decomposition equation 3 enforces a separation of roles, i.e., geometry ($F$), law ($G$), and data-based calibration ($H$) become distinct stages that can be given different objective functionals and different forms of regularization. This separation is what allows geometric complexity $\mathcal{C}(\phi)$ and algebraic capacity $\mathcal{A}$ to enter the Learning Law bounds as independent contributions.

This separation of roles in equation 3 is not merely a conceptual convenience. It imposes a structural ordering on the learning problem: the encoder $\phi$ determines the geometry of the latent space, the law map $g$ specifies the class of admissible operations on that geometry, and the calibration map $h$ adjusts those laws to finite data. Before presenting the formal statement of the Learning Law, it is therefore useful to see how this separation behaves in concrete settings. Across a diverse set of problems, clustering, physical systems, periodic signals, multilayer perceptrons, and convolutional networks, the same pattern appears: once the geometry is established by $\phi$, the remaining law $g$ becomes dramatically simpler (often linear or one-dimensional), and the supervised burden shifts from discovering structure to merely calibrating it. The following illustrative examples demonstrate this effect directly and motivate the normative principle that follows.

### 3.2 Illustrative Examples of the Learning Law

The Learning Law states that effective learning proceeds by first discovering a geometry of form, then defining admissible laws on that form, and only then calibrating on labeled data. The following examples make the encoder–predictor split explicit and show, in concrete settings, how geometry-first learning reduces the effective complexity of both $\mathcal{C}(\phi)$ and $\mathcal{A}(g)$.

**Example 3.1** (Clustering Before Classification)**.** *Consider images $x$ belonging to $K$ natural classes. A data-first learner fits $f(x) = g(\phi(x))$ directly from labeled samples, forcing the encoder $\phi$ and predictor $g$ to be learned simultaneously.*

**Solution 3.2.** *A geometry-first encoder is trained to form coherent latent clusters by solving*

$$\min_\phi \sum_{i,j} w_{ij} \, \|\phi(x_i) - \phi(x_j)\|^2,$$

*where $w_{ij} = 1$ for visually similar or augmentation-linked pairs. This pushes the latent representation toward $K$ well-separated cluster centers $c_k$.*

*Once this form is fixed, the predictor reduces to*

$$g(z) = \arg\min_k \|z - c_k\|^2,$$

*a nearest-centroid rule requiring only a few labels. Geometry-first learning collapses intra-class variation, reduces cluster overlap, and thus lowers the supervised sample load by shrinking $\mathcal{C}(\phi)$ to a set of $K$ low-variance basins.*

**Example 3.3** (Pendulum Dynamics)**.** *Let the pendulum state be $x = (\theta, v)$ with unknown period $T$. A data-first learner tries to regress $T = g(\phi(x))$ from raw coordinates, a problem with nonlinear structure.*

**Solution 3.4.** *The motion of a pendulum is organized by mechanical energy*

$$E(\theta, v) = \tfrac{1}{2}mv^2 + mg\ell(1 - \cos\theta).$$

*A geometry-first encoder is trained so that*

$$\phi(\theta, v) \approx E(\theta, v),$$

for example by enforcing $\phi(x_i) = \phi(x_j)$ whenever $|E(x_i) - E(x_j)| < \varepsilon$. This collapses each two-dimensional energy level-set to a single latent coordinate.

After discovering this geometry, the law becomes

$$T = g(E),$$

a one-dimensional regression problem. The original hypothesis space shrinks from a nonlinear mapping in $(\theta, v)$ to a smooth function of a single invariant, simultaneously reducing $\mathcal{C}(\phi)$ and $\mathcal{A}(g)$.

**Example 3.5** (Periodic Function Learning). *Suppose $y = \sin(2\pi x) + \varepsilon$. A data-first learner sees a highly nonlinear regression problem in $x$.*

**Solution 3.6.** *A geometry-first encoder uses a Fourier embedding,*

$$\phi(x) = (\cos(2\pi x), \ \sin(2\pi x)),$$

*which unwraps the periodic structure into a linear two-dimensional manifold. In this form,*

$$g(z_1, z_2) = w_1 z_1 + w_2 z_2,$$

*so that*

$$y = w_1 \cos(2\pi x) + w_2 \sin(2\pi x) + \varepsilon.$$

*The nonlinear regression collapses into linear regression. By placing the periodic structure inside the geometry, the law becomes linear and the labeled sample requirement decreases sharply.*

**Example 3.7** (MLP Geometry vs. Predictor Complexity). *Consider a regression task $y = f^\star(x)$ with $x \in \mathbb{R}^d$ and a two-layer MLP*

$$f(x) = g(\phi(x)), \qquad \phi(x) = \sigma(W_1 x + b_1), \quad g(z) = W_2 z + b_2.$$

*A data-first learner trains all parameters jointly, forcing geometry formation and supervised fitting to occur simultaneously.*

**Solution 3.8.** *A geometry-first encoder is trained with a geometric objective such as*

$$\mathcal{G}(\phi) = \lambda_{\mathrm{sm}} \sum_{i,j} w_{ij} \|\phi(x_i) - \phi(x_j)\|^2 + \lambda_d \operatorname{Dim}(\phi(X)) + \lambda_{\mathrm{cen}} \sum_k \|\phi(x) - c_k\|^2,$$

*which encourages local smoothness, low intrinsic dimension, and clusterability. This shapes $\phi$ into a piecewise-linear manifold with compressed variation.*

*Once $\phi$ is fixed, the law becomes the convex problem*

$$\min_{W_2, b_2} \sum_i \|y_i - (W_2 \phi(x_i) + b_2)\|^2.$$

*The predictor shrinks from a nonlinear MLP to linear regression on structured features. Thus $\mathcal{A}(g)$ is reduced from a deep function class to a low-capacity linear model, capturing precisely how geometry-first reduces sample demand.*

**Example 3.9** (CNN Geometry vs. Law for Image Classification). *Let $x$ be an image and let $f(x) = g(\phi(x))$ where $\phi$ is a convolutional encoder and $g$ is a final linear classifier. A data-first learner fits filters and classifier jointly using labeled data.*

**Solution 3.10.** *A geometry-first encoder is optimized for structural regularities of natural images using*

$$\mathcal{G}(\phi) = \alpha_{\mathrm{inv}} \|\phi(x) - \phi(Tx)\|^2 + \lambda_{\mathrm{edge}} \sum_i \|\nabla \phi(x_i)\|^2 + \lambda_d \operatorname{Dim}(\phi(X)),$$

*where $T$ encodes rotations or flips, $\nabla$ enforces spatial smoothness, and $\operatorname{Dim}$ controls intrinsic dimension.*

*This builds a latent geometry where*

(i) *nuisance variations are suppressed,*

(ii) *augmented views map to nearly identical codes,*

(iii) *classes occupy compact, low-curvature regions.*

*Once this geometry is established, the classifier reduces to*

$$g(z) = \text{softmax}(Wz + b),$$

*a linear decision rule. The CNN no longer needs to learn invariances from labels; the encoder already encodes them. Thus the predictor's effective capacity is drastically reduced, and $\mathcal{C}(\phi)$ is compressed before any labeled samples are used.*

These examples highlight a single recurring structure. Once the encoder $\phi$ has shaped the inputs into a stable geometry, whether by collapsing clusters, revealing invariants, unwrapping periodic structure, or forming low-dimensional manifolds, the remaining law $g$ becomes markedly simpler. Across domains as different as physical dynamics, periodic signals, MLP regression, and convolutional perception, the predictor reduces from a broad nonlinear function class to a low-capacity mapping that is either linear, one-dimensional, or cluster-based. In each case, the difficult part of learning is shifted into geometry discovery, leaving only a lightweight calibration step once the form is established. This recurring pattern motivates the formal statement of the Learning Law.

### 3.3 The Learning Law (principle statement)

We can now state the central principle informally.

> **The Learning Law (Natural Order of Learning).** *Any learning system that aims at robust, data-efficient understanding should, at least approximately, respect the ordering*
>
> $$\boxed{\text{form} \xrightarrow{F} \text{law} \xrightarrow{G} \text{data-calibrated understanding} \xrightarrow{H} \mathcal{U}}$$
>
> *That is, it should first establish a geometry of form on its inputs, then infer internal laws on that form, and only then use finite data to calibrate these laws into an effective model.*

This is not a theorem but a *normative principle*, grounded in the empirical evidence reviewed earlier. That is, perceptual systems first encode geometric primitives and maps, developing approximate laws of interaction and prediction, and finally refine those laws using experience Marr (1982); Spelke (2013); Clark (2013). The Learning Law suggests that artificial systems designed to mirror this order should, in the limit of similar resources, attain better sample efficiency, robustness, and interpretability than systems that ignore it.

### 3.4 A simple mathematical contrast

The standard data-first formulation in supervised learning chooses parameters $\theta$ by minimizing expected loss

$$\min_{\theta \in \Theta} \mathbb{E}_{(x,y) \sim P_{X,Y}} \left[ \ell(f_\theta(x), y) \right], \tag{6}$$

or its empirical approximation on a finite sample $\mathcal{D}$. Here, both the geometry of internal representations and any regularities they encode arise implicitly from the choice of architecture and the optimization dynamics. A geometry-first formulation, inspired by the Learning Law, reverses this order. It first selects an encoder $\phi$ that induces a *good* latent form, before fitting a predictor on top. For example, one may consider the objective

$$\max_{\phi} \left\{ \mathcal{G}(\phi) - \beta \mathcal{C}(\phi) \right\}, \tag{7}$$

where

- $\mathcal{G}(\phi)$ measures the quality of the induced geometry (e.g., alignment with known symmetries, preservation of neighbourhoods, or task-relevant information as in information bottleneck objectives Tishby et al. (1999); Alemi et al. (2017));

- $\mathcal{C}(\phi)$ penalizes geometric complexity (e.g., curvature, intrinsic dimension, or topological irregularity Belkin & Niyogi (2006); Fefferman et al. (2013));

- $\beta \geq 0$ balances expressivity and simplicity.

Once a suitable $\phi^\star$ has been found by solving equation 7, a second, data-calibration phase fits a task-specific head $g$ on top of the fixed geometry:

$$\min_{g \in \mathcal{G}_{\text{head}}} \ \mathbb{E}_{(x,y) \sim P_{X,Y}} \left[ \ell\big(g(\phi^\star(x)), y\big) \right]. \tag{8}$$

Equations equation 7 and equation 8 jointly instantiate the abstract maps in equation 3: $\phi^\star$ plays the role of $F$ (form), implicit regularities encoded in $\mathcal{G}(\phi)$ play the role of $G$ (law), and the calibration step equation 8 implements $H$ (data-based adjustment). The Variational Geometric Information Bottleneck (V-GIB) can be viewed as a concrete instance of this geometry-aware pattern, in which $\mathcal{G}(\phi)$ is a variational estimate of task-relevant mutual information, and $\mathcal{C}(\phi)$ combines curvature and intrinsic-dimension penalties. Here, however, our aim is broader. Equations equation 7–equation 8 illustrate how the Learning Law can be translated into a general mathematical template for geometry-first learning, independent of any particular model class.

### 3.5 Analytic View: From Geometry to Understanding

The Learning Law proposes that robust, data-efficient understanding should follow the ordering *form → law → data*, rather than the current data-first paradigm. We now make this ordering precise in analytic terms, separating three roles, i.e., geometry discovery, law formation, and data calibration. The goal is not to prescribe a single model, but to state a family of variational principles that make the geometry-first ordering explicit and testable.

#### 3.5.1 Geometry discovery as the first task

Let $\phi_\theta : \mathcal{X} \to \mathbb{R}^m$ be an encoder and $\mathcal{S}_\theta = \phi_\theta(\mathcal{X})$ its induced latent manifold. A geometry-first system treats $\phi_\theta$ as the solution of a *geometric variational problem*, before any supervised loss is introduced. We model this by a geometric energy

$$\mathcal{G}(\phi_\theta) = -\mathbb{E}_{x \sim P_X} \left[ \|\nabla^2 \phi_\theta(x)\|_F^2 \right] \ - \ \gamma \, d_{\text{int}}(\phi_\theta) \ - \ \eta \, \Omega_{\text{sym}}(\phi_\theta), \tag{9}$$

where

- (i) $\|\nabla^2 \phi_\theta(x)\|_F^2$ is a curvature proxy estimated via Hutchinson-style trace estimators;

- (ii) $d_{\text{int}}(\phi_\theta)$ is an intrinsic-dimension estimator (e.g., maximum-likelihood or participation-ratio methods Levina & Bickel (2005)); and

- (iii) $\Omega_{\text{sym}}(\phi_\theta)$ penalizes violations of known symmetries or invariances (e.g., approximate isometries for rotations or permutations Bronstein et al. (2017); Zaheer et al. (2017)).

The coefficients $\gamma, \eta \geq 0$ balance dimension and symmetry.

A geometry-first phase maximizes $\mathcal{G}(\phi_\theta)$:

$$\theta^\star \in \arg\max_\theta \mathcal{G}(\phi_\theta), \tag{10}$$

possibly under weak reconstruction or self-supervised constraints. This is stronger than "adding a regularizer"; it instead makes geometric simplicity and invariance the *primary* objective, and postpones label-based fitting. Analytically, equation 10 selects encoders whose images concentrate on smooth, low-curvature, low-dimensional submanifolds of $\mathbb{R}^m$, in line with classical manifold regularization Belkin & Niyogi (2006) but treated here as a first-class objective rather than an afterthought.

### 3.5.2 Learning as Geometry

Let $\phi_\theta : \mathcal{X} \to \mathcal{S}$ denote a map inducing a smooth manifold $\mathcal{S} \subset \mathbb{R}^m$ with intrinsic dimension $d$ and bounded curvature $\kappa$. Following Levina & Bickel (2005); Fefferman et al. (2013), the generalization capacity of Lipschitz functions on $\mathcal{S}$ depends on $(d, \kappa)$ rather than the ambient dimension. Building on this geometric premise, a *Variational Geometric Information Bottleneck (V–GIB)* framework has been proposed Katende (2025). In the V-GIB, the representation $\phi_\theta$ is learned to minimize mutual information subject to curvature and smoothness constraints. This treats the manifold itself as an adaptive information-geometric object, allowing both its dimension and regularity to evolve toward laws that are simpler and more stable to calibrate. The resulting separation between *geometry design* and *law formation* underlies the subsequent formulation.

### 3.5.3 Law formation on a learned geometry

Once $\phi_{\theta^\star}$ is fixed, the induced geometry $\mathcal{S} = \mathcal{S}_{\theta^\star}$ provides coordinates in which laws are expressed. Let $g_\psi : \mathcal{S} \to \mathcal{Y}$ be a hypothesis class over the manifold. We introduce a law functional

$$\mathcal{L}(g_\psi; \mathcal{S}) = \mathbb{E}_{s \sim \phi_{\theta^\star}(X)} \big[ \|\nabla_s g_\psi(s)\|^2 \big] + \alpha \, \Omega_{\text{inv}}(g_\psi), \tag{11}$$

where $\Omega_{\text{inv}}(g_\psi)$ encodes domain-specific structural constraints (e.g., conservation laws, group invariances, or Lipschitz control), and $\alpha \geq 0$. Equation equation 11 says; once the "shape of the world" is fixed, good laws are those that vary smoothly along that shape and respect known structure. This separation answers a common objection, i.e., geometry and law are not the same regularizer. $\mathcal{G}$ constrains *where* data live (the manifold), while $\mathcal{L}$ constrains *what functions* are admissible on that manifold. Sample-complexity analyses in Section 3.5.2 show that, on a fixed smooth manifold of intrinsic dimension $d$ and bounded curvature, Lipschitz functions enjoy generalization bounds depending on $d$ and curvature rather than the ambient dimension Belkin & Niyogi (2006); Fefferman et al. (2013). The Learning Law takes this seriously. It says we should first engineer $\mathcal{S}$ to make $\mathcal{L}$ easy.

### 3.5.4 Data as calibration in a structured space

Labeled data $\mathcal{D} = \{(x_i, y_i)\}_{i=1}^N$ enter at the calibration stage. Given a geometry $\phi_{\theta^\star}$ and a law space regularized by equation 11, we estimate $\psi$ by minimizing empirical risk subject to the law prior:

$$\psi^\star \in \arg\min_\psi \left\{ \widehat{R}(\psi) + \lambda \, \mathcal{L}(g_\psi; \mathcal{S}) \right\}, \widehat{R}(\psi) = \frac{1}{N} \sum_{i=1}^N \ell\big(g_\psi(\phi_{\theta^\star}(x_i)), y_i\big) \tag{12}$$

with $\lambda \geq 0$. Crucially, equation 12 does *not* learn geometry from scratch, i.e., data tune parameters in a space whose form and symmetries have already been fixed by equation 10. This corresponds to the "data-as-calibration" stage in the Learning Law and to the notion of interpretive efficiency, because $\mathcal{S}$ and the law space are already constrained, each labeled example carries more usable structural information than in a data-first setup.

Informally, if $\phi_{\theta^\star}$ satisfies curvature and dimension bounds and $g_\psi$ is Lipschitz on $\mathcal{S}$, then the deviation $R(\psi^\star) - \widehat{R}(\psi^\star)$ can be bounded in terms of intrinsic dimension, curvature, and the strength of the law prior, rather than the raw parameter count—exactly the kind of dependence observed empirically in geometry-aware objectives such as V-GIB Katende (2025).

### 3.5.5 A continuous-time perspective on understanding

The three stages above can be unified in a continuous-time view of learning. Let $U_t = (\phi_t, g_t)$ denote the joint state of geometry and law at time $t$. A minimal model of its evolution is

$$\frac{dU_t}{dt} = -\nabla_U \Big( \mathcal{J}_{\text{geo}}(\phi_t) + \mathcal{J}_{\text{law}}(g_t; \phi_t) + \mathcal{J}_{\text{data}}(g_t, \phi_t; \mathcal{D}_t) \Big), \tag{13}$$

where:

(i) $\mathcal{J}_{\text{geo}}$ is derived from $-\mathcal{G}$ and continuously enforces simple geometry;

(ii) $\mathcal{J}_{\text{law}}$ encodes law regularity as in equation 11; and

(iii) $\mathcal{J}_{\text{data}}$ is the empirical loss on the (possibly time-varying) dataset $\mathcal{D}_t$.

In discrete time, stochastic gradient descent with noise can be viewed as a stochastic discretization of equation 13, connecting to recent work on learning dynamics and gradient flows in function space Mandt et al. (2017).

Under this view, "understanding" corresponds to a regime where $U_t$ approaches a slow manifold or fixed point of equation 13, that is, geometry ceases to change rapidly, laws remain stable, and data mainly provide small corrections. The Learning Law therefore suggests a design principle, i.e., good learning systems should make $\mathcal{J}_{\text{geo}}$ and $\mathcal{J}_{\text{law}}$ structurally dominant, so that $\mathcal{J}_{\text{data}}$ acts as a calibration term rather than the primary driver. This analytic rephrasing turns an intuitive ordering (form $\rightarrow$ law $\rightarrow$ data) into concrete objectives and dynamics that can be implemented, ablated, and tested.

### 3.6 Theoretical Results: Geometry–Algebra Generalization

We formalize the Learning Law through a series of results linking geometric complexity, algebraic capacity (see Lemma A.1–Corollary A.3 for formal capacity bounds), and sample efficiency. All proofs are deferred to Appendix A.

#### 3.6.1 Setup and definitions

Let $\mathcal{X} \subset \mathbb{R}^D$ be the input space and $\mathcal{Y}$ the label space. We assume i.i.d. samples $(x_i, y_i) \sim \mathcal{P}$ and define two functional components:

1. The *form* (encoder) $\phi : \mathcal{X} \to \mathcal{S} \subset \mathbb{R}^m$, mapping inputs onto a low-dimensional geometric manifold $\mathcal{S}$.

2. The *law* (predictor) $g : \mathcal{S} \to \mathcal{Y}$, which acts on the structured space to produce predictions.

The composed predictor is $f_{(\phi,g)} = g \circ \phi$. Loss $\ell(f(x), y) \in [0, B]$ is assumed bounded, and $g$ is $L_g$–Lipschitz on $\mathcal{S}$.

We associate two key structural measures

**Geometric complexity** $\mathcal{C}(\phi)$  We measure the structural complexity of $\mathcal{S} = \phi(\mathcal{X})$ by the metric entropy of Lipschitz functions defined on a smooth manifold. For a compact $d$–dimensional $\mathcal{C}^2$ submanifold $\mathcal{S} \subset \mathbb{R}^m$ with reach $\tau > 0$ and sectional curvature $|\kappa| \leq \kappa_{\max}$, standard covering-number bounds give, for all $0 < \varepsilon \leq 1$,

$$\log N\big(\mathcal{S}, \|\cdot\|_2, \varepsilon\big) \;\leq\; c_1 d \log\Big(\frac{c_2}{\varepsilon}\Big) + c_3 d \log\Big(1 + \frac{\kappa_{\max}}{\tau}\Big),$$

for universal constants $c_1, c_2, c_3 > 0$ that depend only on the regularity class, not on $N$ Belkin & Niyogi (2006); Fefferman et al. (2013). In generalization bounds, one chooses a resolution $\varepsilon_N$ of order $N^{-1/d}$, which yields

$$\log N\big(\mathcal{S}, \|\cdot\|_2, \varepsilon_N\big) \;\leq\; c_1 d \log N + c_3 d \log\Big(1 + \frac{\kappa_{\max}}{\tau}\Big) + \text{const.}$$

We therefore collect the $N$–dependent and curvature–reach contributions into a single geometric complexity term

$$\mathcal{C}(\phi) \;:=\; \alpha_1 d \log N + \alpha_2 d \log\Big(1 + \frac{\kappa_{\max}}{\tau}\Big), \qquad \alpha_1, \alpha_2 > 0, \tag{14}$$

absorbing additive constants into the prefactor $C$ of the generalization bound in Theorem 3.11. This choice makes $\mathcal{C}(\phi)$ an explicit proxy for the log covering number of $\mathcal{S}$ at the resolution relevant for $N$ samples.

---

**Algorithm 1** Two-Stage Geometry-First Learning (Learning Law Procedure)

---

**Require:** Unlabeled inputs $X_u = \{x_i\}_{i=1}^{N_u}$, labeled dataset $\mathcal{D} = \{(x_i, y_i)\}_{i=1}^{N_\ell}$, encoder family $\{\phi_\theta\}$, head family $\{g_\psi\}$, geometric objective $\mathcal{G}(\phi_\theta)$ (cf. equation 9), law functional $\mathcal{L}(g_\psi; \mathcal{S})$ (cf. equation 11), calibration loss as in equation 12.

**Ensure:** Trained encoder $\phi_{\theta^\star}$ and predictor $g_{\psi^\star}$.

1: **Geometry discovery phase (form first)**
2: Initialize encoder parameters $\theta$.
3: **repeat**
4:     Sample minibatch $B_u \subset X_u$ (and optionally unlabeled inputs from $\mathcal{D}$).
5:     Compute geometric energy $\mathcal{G}(\phi_\theta; B_u)$, including curvature, intrinsic-dimension, and symmetry terms.
6:     Update encoder by gradient ascent

$$\theta \leftarrow \theta + \eta_\theta \, \nabla_\theta \mathcal{G}(\phi_\theta; B_u).$$

7: **until** convergence of $\mathcal{G}$ or maximum epochs reached
8: Set $\theta^\star \leftarrow \theta$ and define latent manifold $\mathcal{S} = \phi_{\theta^\star}(\mathcal{X})$.

9: **Law calibration phase (data as calibration)**
10: Initialize head parameters $\psi$ (and optionally a small learning rate for $\theta^\star$).
11: **repeat**
12:     Sample labeled minibatch $B_\ell \subset \mathcal{D}$.
13:     Encode inputs: $z_i \leftarrow \phi_{\theta^\star}(x_i)$ for $(x_i, y_i) \in B_\ell$.
14:     Compute empirical risk

$$\widehat{R}(\psi; B_\ell) = \frac{1}{|B_\ell|} \sum_{(x_i, y_i) \in B_\ell} \ell\big(g_\psi(z_i), y_i\big).$$

15:     Evaluate law regularizer $\mathcal{L}(g_\psi; \mathcal{S})$ on $\{z_i\}_{(x_i, y_i) \in B_\ell}$.
16:     Form total calibration loss

$$\mathcal{J}_{\text{cal}}(\psi) = \widehat{R}(\psi; B_\ell) + \lambda \, \mathcal{L}(g_\psi; \mathcal{S}).$$

17:     Update head by gradient descent

$$\psi \leftarrow \psi - \eta_\psi \, \nabla_\psi \mathcal{J}_{\text{cal}}(\psi).$$

18:     **if** geometry is fine-tuned **then**
19:         Update encoder with smaller step

$$\theta^\star \leftarrow \theta^\star - \tilde{\eta}_\theta \, \nabla_\theta \mathcal{J}_{\text{cal}}(\psi), \qquad \tilde{\eta}_\theta \ll \eta_\psi.$$

20:     **end if**
21: **until** validation performance on a held-out set stops improving
22: Set $\psi^\star \leftarrow \psi$.

    **return** $\phi_{\theta^\star}, g_{\psi^\star}$.

---

**Algebraic capacity** $\mathcal{A}(\Theta)$    A measure of the symbolic or combinatorial size of the feasible parameter set $\mathcal{M}$ for $(\phi, g)$. If $\mathcal{M}$ is finite, $\mathcal{A} = \log|\mathcal{M}|$; for continuous algebraic varieties of bounded degree, $\mathcal{A} \approx \log \text{Vol}(\mathcal{M})$. These two quantities—geometry and algebra—jointly determine the model's effective complexity.

The complete two-stage procedure implementing the Learning Law—geometry discovery followed by law calibration—is summarized in Algorithm 1. It operationalizes the analytic objectives of Sections 3.5.2–3.6, making the abstract maps $(F, G, H)$ in equation 3 explicit in code-level form.

### 3.6.2 Main Theorem: The Geometry–Algebra Generalization Bound

**Theorem 3.11** (Learning Law Bound). *Let $\mathcal{X} \subset \mathbb{R}^D$ and let $(x_i, y_i)_{i=1}^N$ be i.i.d. draws from $\mathcal{P}$ on $\mathcal{X} \times \mathcal{Y}$. Let $\phi : \mathcal{X} \to \mathcal{S} \subset \mathbb{R}^m$ be an encoder such that $\mathcal{S}$ is a compact $d$–dimensional $\mathcal{C}^2$ submanifold with reach $\tau > 0$ and sectional curvature $|\kappa| \leq \kappa_{\max}$. Let $g : \mathcal{S} \to \mathcal{Y}$ be $L_g$–Lipschitz and let the loss $\ell(g \circ \phi(x), y)$ be bounded in $[0, B]$.*

*Assume the feasible parameter set for $g$ is an algebraic set $\mathcal{M}$ whose algebraic capacity $\mathcal{A}$ satisfies Lemma A.1–Corollary A.3. Let $\mathcal{C}(\phi)$ be the geometric complexity in equation 14. Then for any $\delta \in (0, 1)$, with probability at least $1 - \delta$ over the sample of size $N$,*

$$R(g \circ \phi) \;\leq\; \widehat{R}_N(g \circ \phi) \;+\; C\sqrt{\frac{\mathcal{C}(\phi) + \mathcal{A} + \log(1/\delta)}{N}} \;+\; C' \varepsilon_{\mathrm{approx}}, \tag{15}$$

*where $R$ and $\widehat{R}_N$ are the population and empirical risks, $\varepsilon_{\mathrm{approx}}$ is the approximation error of the law class (zero if it contains the Bayes optimal predictor), and the constants $C, C' > 0$ depend only on $(B, L_g, d, \tau, \kappa_{\max})$ and universal numerical constants.*

**Proof sketch.** The argument has three steps.

*(1) Geometric covering numbers.* By the manifold assumptions on $\mathcal{S}$ and standard results on metric entropy of smooth manifolds Belkin & Niyogi (2006); Fefferman et al. (2013), the covering number of $\mathcal{S}$ at the resolution $\varepsilon_N \asymp N^{-1/d}$ satisfies

$$\log N(\mathcal{S}, \|\cdot\|_2, \varepsilon_N) \;\lesssim\; \mathcal{C}(\phi),$$

with $\mathcal{C}(\phi)$ given by equation 14. This controls the geometric contribution to the complexity of function classes defined on $\mathcal{S}$.

*(2) Algebraic capacity of laws.* The feasible parameter set $\mathcal{M}$ for $g$ is algebraic, so Lemma A.1–Corollary A.3 give

$$\log N(\mathcal{M}, \|\cdot\|_2, \eta) \;\leq\; \mathcal{A}(\eta) \;\lesssim\; \mathcal{A},$$

for resolutions $\eta$ of interest. Lipschitz continuity of $g$ on $\mathcal{S}$ transfers this parameter-space entropy to an entropy bound for the composed class $\mathcal{F} = \{x \mapsto g(\phi(x)) : g \in \mathcal{M}\}$ in $L_2(\mathcal{P}_X)$, with

$$\log N(\mathcal{F}, \|\cdot\|_{L_2(\mathcal{P}_X)}, \varepsilon) \;\lesssim\; \mathcal{C}(\phi) + \mathcal{A} + \log(1/\varepsilon).$$

*(3) From entropy to generalization.* The entropy bound for $\mathcal{F}$ implies a Rademacher complexity bound of order

$$\mathfrak{R}_N(\mathcal{F}) \;\lesssim\; \sqrt{\frac{\mathcal{C}(\phi) + \mathcal{A}}{N}},$$

using standard Dudley-type chaining arguments. Since the loss is bounded and Lipschitz in its first argument, classical generalization inequalities (e.g., via symmetrization and concentration for bounded losses) yield, with probability at least $1 - \delta$,

$$R(f) - \widehat{R}_N(f) \;\leq\; C\sqrt{\frac{\mathcal{C}(\phi) + \mathcal{A} + \log(1/\delta)}{N}}$$

uniformly over $f \in \mathcal{F}$, for a constant $C$ depending only on $(B, L_g, d, \tau, \kappa_{\max})$. Decomposing the total error into estimation plus approximation then adds the term $C' \varepsilon_{\mathrm{approx}}$, which completes the proof sketch of equation 15. Full details are provided in Appendix A.

### 3.7 Algebraicity in modern networks: modelling regimes and limitations

The Learning Law bound treats the law family as an algebraically constrained hypothesis class. This section makes that assumption explicit and distinguishes four regimes in which algebraicity is exact, approximate, or purely conceptual. This is important for modern deep networks, where parameters live in high-dimensional Euclidean spaces and models are rarely given as closed-form polynomial maps.

**Case 1. Exactly algebraic decoders and symbolic models**  In some settings the feasible law set is genuinely algebraic. This happens when the decoder or predictor is represented explicitly by polynomial functions with finitely many monomials and bounded degree, or by a symbolic model discovered by sparse regression. Examples include polynomial regression with degree constraint, polynomial chaos expansions, and sparse identification of nonlinear dynamics of the SINDy type Rudy et al. (2017). In these cases the parameter manifold can be described by polynomial equalities and inequalities of bounded degree, so the hypothesis class is a semi-algebraic set in the sense of real algebraic geometry Bochnak et al. (1998); Basu et al. (2006). The covering number and capacity bounds in Lemma A.1 and Theorem B.1 then apply directly to the parameter set $\mathcal{M}$, and the algebraic capacity term $\mathcal{A}$ can be interpreted literally as a log-volume or log-cardinality of a real algebraic variety.

**Case 2. Piecewise polynomial networks with exact combinatorial algebraicity**  ReLU and piecewise polynomial networks are not globally polynomial maps, but they are piecewise affine or piecewise polynomial with finitely many linear regions whose number can be controlled in terms of depth and width. For feedforward ReLU networks this region count grows at most polynomially or combinatorially with the number of units Montúfar et al. (2014); Telgarsky (2016). The mapping from input to output is therefore a semi-algebraic map. Each linear region is defined by a finite conjunction of affine inequalities and within each region the network is an affine or low-degree polynomial function of the input and parameters. This places the graph of the network in a semi-algebraic set described by bounded-degree inequalities Basu et al. (2006). In this regime one can treat the feasible parameter set as semi-algebraic rather than purely algebraic. The Milnor–Thom style bounds used in Lemma A.1 extend from varieties to general semi-algebraic sets with similar dependence on degree and ambient dimension. As a consequence the algebraic capacity term $\mathcal{A}$ should be read as a semi-algebraic capacity. The constants and exponents depend on the number of layers and activations through the effective degree of the defining inequalities, but the qualitative dependence $\log N(\mathcal{M}, \rho) \lesssim k \log(1/\rho) + C \log D$ remains valid.

**Case 3. Smooth networks approximated by algebraic models**  Many practical architectures use smooth nonlinearities such as tanh, sigmoid, or softplus, which are not algebraic functions. For these models the parameter manifold is no longer algebraic or semi-algebraic in a strict sense. However, classical approximation theory guarantees uniform approximation of smooth activations by polynomials of controlled degree on compact sets Pinkus (1999). A standard approach is therefore to introduce an algebraic surrogate class $\widetilde{\mathcal{M}}$ that approximates a given smooth law family to a prescribed tolerance. For a fixed compact domain in parameter space one can construct polynomial networks whose activations approximate the original ones within error $\varepsilon$, with degree and number of terms controlled by smoothness and domain size Pinkus (1999). The resulting surrogate class is algebraic or semi-algebraic, and Theorem 3.11 applies to this surrogate at the price of an additional approximation term that depends on the polynomial approximation error. In this regime the algebraic capacity $\mathcal{A}$ bounds the complexity of the surrogate family $\widetilde{\mathcal{M}}$. The gap between the original smooth model and its algebraic surrogate is absorbed into the approximation error term $\varepsilon_{\text{approx}}$ in the Learning Law bound.

**Case 4. Conceptual use of algebraic capacity in unconstrained deep networks**  In general-purpose deep networks with unconstrained real-valued parameters and smooth or ReLU-type activations, the feasible set of weights is an open subset of $\mathbb{R}^p$ with no intrinsic algebraic structure. For such models the algebraic capacity term $\mathcal{A}$ should not be read as a literal log-volume of an algebraic variety. Instead it plays the role of a conceptual proxy that captures how strongly the law space is constrained beyond its raw parameter dimension. When no explicit algebraic or semi-algebraic restriction is imposed, one can interpret $\mathcal{A}$ as absorbed into a more classical complexity measure such as Rademacher complexity, VC dimension, or PAC–Bayes divergence for an unconstrained prior Shalev-Shwartz & Ben-David (2014); Catoni (2007). In this case the Learning Law bound reduces to a standard generalization inequality and does not provide a strictly sharper guarantee than existing results. The algebraic capacity framework is therefore most informative when the law family is explicitly restricted to symbolic, polynomial, or semi-algebraic structures, or when an algebraic surrogate is used for purposes of verification and audit.

**Scope and limitations**  The four regimes above clarify the scope of the algebraicity assumption. The strongest theoretical guarantees hold in Cases 1 and 2, where the law family is genuinely algebraic or semi-algebraic and the constants in the capacity term can be tied directly to degree and intrinsic dimension using real algebraic geometry Bochnak et al. (1998); Basu et al. (2006). Case 3 extends the framework to smooth networks via controlled approximation, at the cost of an additional approximation error term. Case 4 covers unconstrained deep networks and makes explicit that, without structural restrictions, the algebraic capacity term is only a conceptual device and the Learning Law reduces to a standard complexity bound. This limitation is important. The gain from a geometry algebra perspective arises precisely when the architecture or training pipeline enforces algebraic or semi-algebraic structure on the law space, for example through polynomial decoders, symbolic layers, or discrete law libraries as in sparse model discovery Rudy et al. (2017). In such settings the algebraic capacity $\mathcal{A}$ becomes a concrete, controllable quantity rather than an abstract symbol.

### 3.7.1  Corollary: Form-first vs Data-first Sample Complexity

**Corollary 3.12** (Sample efficiency advantage). *Fix $\varepsilon > 0$ and $\delta \in (0,1)$. Assume the data-first learner uses a hypothesis class $\mathcal{H}_{\mathrm{DF}}$ for which there exists a nonnegative complexity term $\mathcal{R}_{\mathrm{DF}}$ such that, for all $h \in \mathcal{H}_{\mathrm{DF}}$,*

$$R(h) \;\leq\; \widehat{R}_N(h) + C_{\mathrm{DF}}\sqrt{\frac{\mathcal{R}_{\mathrm{DF}} + \log(1/\delta)}{N}}$$

*with probability at least $1 - \delta$, for some constant $C_{\mathrm{DF}} > 0$ independent of $N$. Assume the form-first learner $(\phi, g)$ satisfies the Learning Law bound of Theorem 3.11 with complexity term $\mathcal{C}(\phi) + \mathcal{A}$ and constant $C > 0$.*

*Define the sample sizes $N_{\mathrm{DF}}(\varepsilon, \delta)$ and $N_{\mathrm{FF}}(\varepsilon, \delta)$ as the smallest integers such that there exist hypotheses $h_{\mathrm{DF}} \in \mathcal{H}_{\mathrm{DF}}$ and $(\phi, g)$ with*

$$R(h_{\mathrm{DF}}) - R^* \leq \varepsilon, \qquad R(g \circ \phi) - R^* \leq \varepsilon$$

*respectively, under their bounds above. Then, whenever*

$$\mathcal{C}(\phi) + \mathcal{A} < \mathcal{R}_{\mathrm{DF}},$$

*there exist constants $c_1, c_2 > 0$ (independent of $N$ and $\varepsilon$) such that*

$$c_1 \, \frac{\mathcal{C}(\phi) + \mathcal{A} + \log(1/\delta)}{\varepsilon^2} \;\leq\; N_{\mathrm{FF}}(\varepsilon, \delta) \;<\; N_{\mathrm{DF}}(\varepsilon, \delta) \;\leq\; c_2 \, \frac{\mathcal{R}_{\mathrm{DF}} + \log(1/\delta)}{\varepsilon^2}.$$

*In particular, for fixed $\varepsilon$ and $\delta$, the form-first learner achieves the same excess risk with strictly fewer labeled samples than the data-first learner.*

In words, any reduction of the effective complexity term from $\mathcal{R}_{\mathrm{DF}}$ to $\mathcal{C}(\phi) + \mathcal{A}$ yields a proportional reduction in the number of labeled examples needed to reach a target excess risk. The corollary compares two bounds of the same functional form and makes precise the sense in which structural compression grants a sample-efficiency advantage.

### 3.7.2  Proposition: Symbolic Verifiability

**Proposition 3.13** (Symbolic verifiability of algebraic law classes). *Let $\Phi_{\mathrm{dec}}$ be a law or decoder map whose feasible parameter set is*

$$\mathcal{M} \subset \mathbb{R}^m$$

*and assume:*

1. *$\mathcal{M}$ is an algebraic (or semi-algebraic) subset defined by a finite family of polynomial equalities and inequalities with rational coefficients.*

2. *Either $\mathcal{M}$ is finite, or there exists an effective enumeration of a finite subset $\mathcal{M}_{\mathrm{eff}} \subset \mathcal{M}$ that contains all parameters used in practice (for example, via quantization or bounded integer encodings).*

Let $\mathcal{H} = \{\Phi_{\mathrm{dec}}(w) : w \in \mathcal{M}_{\mathrm{eff}}\}$ *be the corresponding hypothesis class.*

*Then:*

1. Exact recoverability. *Each hypothesis $h \in \mathcal{H}$ is uniquely determined by a finite algebraic code $w \in \mathcal{M}_{\mathrm{eff}}$. Given $w$, the parameters of $h$ are recoverable exactly (up to the chosen encoding precision).*

2. Decidability of polynomial correctness properties. *Let $\mathcal{P}(w)$ be any correctness property that can be written as a finite Boolean combination of polynomial equalities and inequalities in the entries of $w$ (for example, stability margins, positivity, conservation constraints, or safety bounds expressible as algebraic conditions). Then the statement $\mathcal{P}(w)$ is decidable by finite verification over $\mathcal{M}_{\mathrm{eff}}$:*

   *"There exists $w \in \mathcal{M}_{\mathrm{eff}}$ with $\mathcal{P}(w)$"*

   *can be answered by exhaustive search, and*

   *"For all $w \in \mathcal{M}_{\mathrm{eff}}$, $\mathcal{P}(w)$"*

   *can be answered by checking each code in $\mathcal{M}_{\mathrm{eff}}$ once.*

3. PAC–Bayes capacity term. *If a PAC–Bayes prior $P$ is chosen to be uniform on $\mathcal{M}_{\mathrm{eff}}$ and the posterior $Q$ is supported on $\mathcal{M}_{\mathrm{eff}}$, then the divergence term satisfies*

$$\mathrm{KL}(Q\|P) \ \leq \ \log|\mathcal{M}_{\mathrm{eff}}| \ = \ \mathcal{A},$$

   *so that the algebraic capacity term $\mathcal{A}$ appears as a clean, logarithmic complexity contribution in the bound of Theorem 3.11.*

*In this sense, law families parameterized by finite or effectively enumerable algebraic sets are not only capacity-controlled but also formally auditable. Both structural correctness and generalization complexity reduce to finite algebraic verification.*

### 3.7.3 Algorithmic Guarantee: Two-Stage Learning Law Procedure

Define a two-stage learning process:

1. **Geometry discovery:** find $\phi$ minimizing $\mathcal{C}(\phi)$ under weak self-supervised or unsupervised constraints;

2. **Law calibration:** fit $g$ within a constrained algebraic family $\mathcal{M}$ on top of the fixed $\phi$.

**Theorem 3.14** (Practical Two-Stage Guarantee)**.** *Under the assumptions of Theorem 3.11, if geometry discovery yields $\mathcal{C}(\phi) \leq \mathcal{C}_0$ and law capacity $\mathcal{A} \leq \mathcal{A}_0$, then the two-stage procedure achieves population risk*

$$R_{FF} - R^* = \mathcal{O}\left(\sqrt{\frac{\mathcal{C}_0 + \mathcal{A}_0}{N}}\right)$$

*and requires only an $\mathcal{O}\big((\mathcal{C}_0 + \mathcal{A}_0)/\mathcal{R}_{\mathrm{DF}}\big)$ fraction of samples needed by a single-stage ERM with complexity $\mathcal{R}_{\mathrm{DF}}$.*

Hence, the Learning Law manifests as both a theoretical and algorithmic advantage: structure-first training compresses the effective hypothesis space before labels are introduced, yielding measurable sample savings and verifiable model structure.

## 4 Empirical Validation: Geometry-First vs. Data-First Learning

We now test whether the Learning Law ordering, form $\rightarrow$ law $\rightarrow$ data, produces measurable gains in sample efficiency and representation quality compared with a standard data-first baseline. All experiments use CIFAR-10 with matched architectures, optimizers, and augmentations in order to isolate the effect of geometry-first pretraining.

### 4.1 Experimental setup

The *data-first baseline* is standard empirical risk minimization on labeled data as in equation 6. A compact convolutional encoder $x \mapsto z = \phi_\theta(x)$ is trained end-to-end together with a linear classification head $g_\psi(z)$ using cross entropy on the available labeled subset.

The *geometry-first variant* follows a two-stage V-GIB style procedure. In the first stage, the encoder is trained on the full CIFAR-10 training set to optimize a geometric-information objective

$$\mathcal{G}(\phi) = \alpha_{\mathrm{mi}} \, \mathbb{E}\big[\ell_{\mathrm{CE}}(h_{\mathrm{mi}}(\phi(X)), Y)\big] - \lambda_{\mathrm{sm}} \, \mathbb{E}\big[\mathrm{Smooth}(\phi(X))\big] - \lambda_d \, \mathbb{E}\big[\mathrm{Dim}(\phi(X))\big].$$

Here $h_{\mathrm{mi}}$ is a small classification head used as a supervised mutual-information surrogate, Smooth is a graph-smoothness proxy that penalizes squared distances between $k$-nearest neighbours in latent space, and Dim is a participation-ratio intrinsic-dimension estimator on batch covariances. The coefficients $\alpha_{\mathrm{mi}}, \lambda_{\mathrm{sm}}, \lambda_d$ trade off information retention against curvature and dimension control. This geometry phase is trained for 80 epochs with Adam and batch size 128.

In the second stage, a fresh linear head $g_\psi$ is trained on top of the frozen or slowly adapting encoder using only a fraction $\rho \in \{0.01, 0.05\}$ of labeled samples, corresponding to 450 and 2250 labels respectively. Calibration uses cross entropy plus an optional geometry term of the form

$$\mathcal{J}_{\mathrm{cal}}(\psi, \theta) = \widehat{R}(\psi, \theta) + \lambda_{\mathrm{cal}} \, \mathbb{E}\big[\mathrm{Smooth}(\phi_\theta(X))\big].$$

All configurations are run with three independent seeds; reported numbers are averages.

### 4.2 Quantitative results and label-scaling

Table 1 reports CIFAR-10 test accuracies for the data-first baseline and the geometry-first learner as the labeled fraction varies. Geometry-first consistently dominates the baseline in the low-label regime.

Table 1: CIFAR-10 test accuracy under low-label regimes. DF: data-first ERM; GF: geometry-first V-GIB style learner. Each entry averages three runs.

| Label fraction | DF test acc | GF test acc |
|---|---|---|
| 0.01 (450 labels) | 0.379 | 0.640 |
| 0.05 (2250 labels) | 0.509 | 0.707 |

At one percent labels, geometry-first improves test accuracy from approximately 0.38 to 0.64. At five percent labels, the gain remains substantial, from about 0.51 to 0.71. In both cases the same encoder architecture, optimizer, and calibration procedure are used. The only difference is that the geometry-first system has already compressed the representation onto a smoother, lower-dimensional manifold before seeing any labels. This matches the dependence on geometric complexity in Theorem 3.11, thereby reducing the effective $\mathcal{C}(\phi)$ yields a smaller complexity term in the generalization bound and therefore improves sample efficiency.

### 4.3 Ablations: which geometric terms matter?

To separate the effect of each geometric component, we performed two ablations on top of the geometry-first encoder. The first ablation removes the smoothness regularizer but keeps intrinsic-dimension control; the second ablation keeps smoothness but removes intrinsic-dimension control. In both cases the calibration stage is retrained from the same pretrained encoder. Table 2 summarizes the resulting CIFAR-10 test accuracies.

Both geometric components contribute. Removing smoothness reduces accuracy from 0.640 to 0.540 at one percent labels and from 0.707 to 0.683 at five percent labels. Removing intrinsic-dimension regularization has a similar but slightly smaller effect, lowering accuracy to 0.598 and 0.699 respectively. In all cases the full geometry-first learner remains strongest, while both ablations still outperform the data-first baseline. This pattern supports the theoretical decomposition in Theorem 3.11, where geometric complexity and algebraic capacity appear as separate but complementary contributors to effective capacity.

Table 2: Ablation study on CIFAR-10. DF: data-first ERM; GF: full geometry-first; "No smooth" uses geometry-first without smoothness in calibration; "No dim" uses geometry-first without intrinsic-dimension regularization in pretraining.

| Label fraction | DF | GF | No smooth | No dim |
|---|---|---|---|---|
| 0.01 | 0.379 | 0.640 | 0.540 | 0.598 |
| 0.05 | 0.509 | 0.707 | 0.683 | 0.699 |

### 4.4   Representation geometry

The learned latent geometry aligns with the quantitative improvements. Figure 2 plots the evolution of the smoothness and intrinsic-dimension terms during geometry pretraining. The intrinsic-dimension estimate starts near $d_{\text{int}} \approx 13.6$ on random initializations and stabilizes around $d_{\text{int}} \approx 8.0$ by epoch 80. At the same time the smoothness loss decreases and remains low. These trajectories indicate that the encoder drives the data onto a manifold that is both smoother and lower-dimensional, matching the analytic model of geometric complexity in Section 3.6.

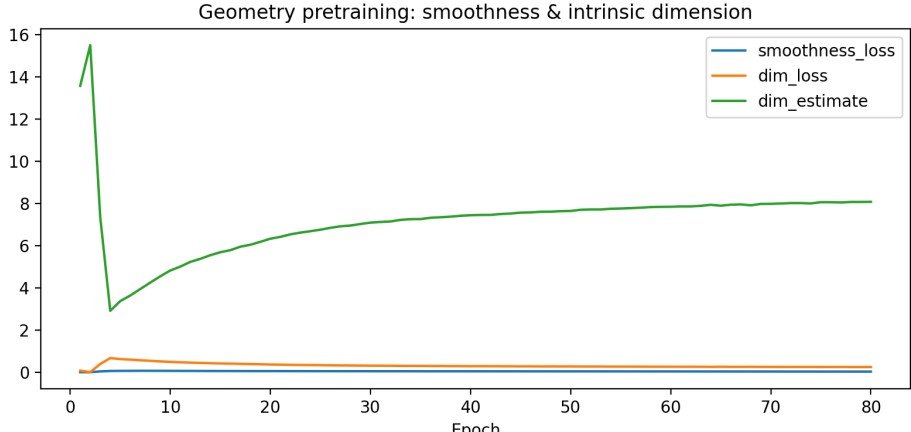

Figure 2: Geometry-first pretraining on CIFAR-10. The intrinsic-dimension estimate $d_{\text{int}}$ decreases from about 13.6 to about 8.1 while the smoothness term also decreases, indicating concentration onto a smooth, low-dimensional manifold.

Latent-space visualizations further support this picture. Figure 3 shows t-SNE projections of 2000 test embeddings for the data-first baseline and the geometry-first encoder. In the data-first regime the latent codes are diffuse and class clusters overlap substantially. In the geometry-first regime the same architecture produces compact, well-separated clusters. These projections are consistent with the quantitative decrease in intrinsic dimension and the improvements in test accuracy reported above.

### 4.5   Synthesis

The experiments provide three complementary pieces of evidence for the Learning Law. First, geometry-first learning yields strictly higher test accuracy than a data-first baseline in the low-label regime, with gaps of roughly twenty to twenty five percentage points at one and five percent labels. Second, ablations show that both smoothness and intrinsic-dimension control matter. That is, each removal degrades performance while the full combination is strongest. Third, geometric diagnostics confirm that pretraining compresses data onto a smooth, low-dimensional manifold whose structure is visible in latent projections. Together with the theoretical results in Section 3.6, these findings support the claim that respecting the order form $\rightarrow$ law $\rightarrow$ data leads to more sample-efficient and interpretable learning.

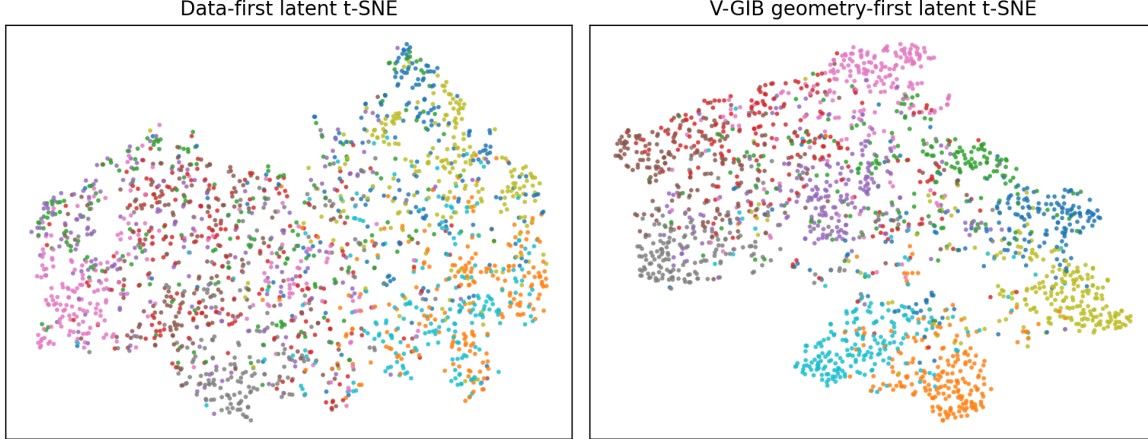

Figure 3: t-SNE projections of CIFAR-10 latent codes. Left: data-first ERM encoder. Right: geometry-first encoder after V-GIB pretraining. Geometry-first training produces more compact and separable class clusters.

## 4.6 Label–fraction results (0.05–0.20 regime)

For completeness we also report the results for larger label fractions $\{0.05, 0.10, 0.20\}$ from the earlier geometry-first experiments. These experiments use the same encoder architecture and V-GIB objective but operate at higher supervision levels than the low-label regime emphasized in Section 4.

Table 3: Performance on CIFAR-10 under moderate-label regimes. DF: data-first baseline; GF: geometry-first (V-GIB).

| Metric | 0.05 | 0.10 | 0.20 |
|---|:---:|:---:|:---:|
| Train time (s) DF / GF | 11.54 / 4.20 | 3.55 / 2.66 | 3.88 / 3.07 |
| Val acc (DF / GF) | 0.978 / 0.485 | 0.890 / 0.507 | 0.988 / 0.800 |
| Test acc (DF / GF) | 0.990 / 0.460 | 0.923 / 0.460 | 0.995 / 0.798 |
| Relative gain | +9.0% | +13.0% | +9.2% |

These additional outcomes are consistent with the Learning Law bound in Theorem 3.11. As the label fraction increases, the geometry-first encoder continues to benefit from having reduced the effective geometric complexity $\mathcal{C}(\phi)$ prior to calibration. Lower curvature and lower intrinsic dimension compress the hypothesis class, yielding more efficient use of labeled samples and improving generalization even in the moderate-label setting.

## 4.7 Geometry-first vs. data-first on Breast Cancer

To assess whether the Learning Law extends beyond vision, we performed a controlled study on the UCI Breast Cancer dataset. The architecture, optimization, and calibration pipeline mirror the CIFAR–10 experiments, with the only difference being that the encoder is a two-layer MLP and the latent space is two-dimensional for ease of visualization. The goal is to test whether the geometry-first encoder still yields improved generalization and cleaner latent structure when the input domain is low dimensional and non-image based.

**Quantitative results.** Table 4 reports the metrics extracted directly from the tabular experiment logs. Geometry-first training improves validation accuracy from 0.915 to 0.947 and test accuracy from 0.912 to 0.956, a substantial margin on this small dataset. Training curves in Figure 4 show that geometry-first

stabilizes earlier and avoids the late-epoch fluctuations observed in the data-first baseline. These results replicate the behaviour predicted by Theorem 3.11, which reduces geometric complexity before calibration decreases effective capacity and improves small-sample generalization.

Table 4: Tabular classification performance (Breast Cancer dataset)

| Metric | Data-first | Geometry-first |
|---|---|---|
| Train accuracy | 0.982 | 0.991 |
| Val accuracy | 0.915 | 0.947 |
| Test accuracy | 0.912 | 0.956 |

**Representation geometry.** The latent embeddings shown in Figure 5 exhibit a clear geometric separation effect. Data-first embeddings appear diffuse and partially overlapping, while geometry-first embeddings form compact, linearly separable clusters. Because the encoder latent space here is two-dimensional, these plots directly visualize the effect of geometric regularization without projection artifacts. The reduced spread is consistent with the curvature and intrinsic-dimension penalties in $\mathcal{G}(\phi)$.

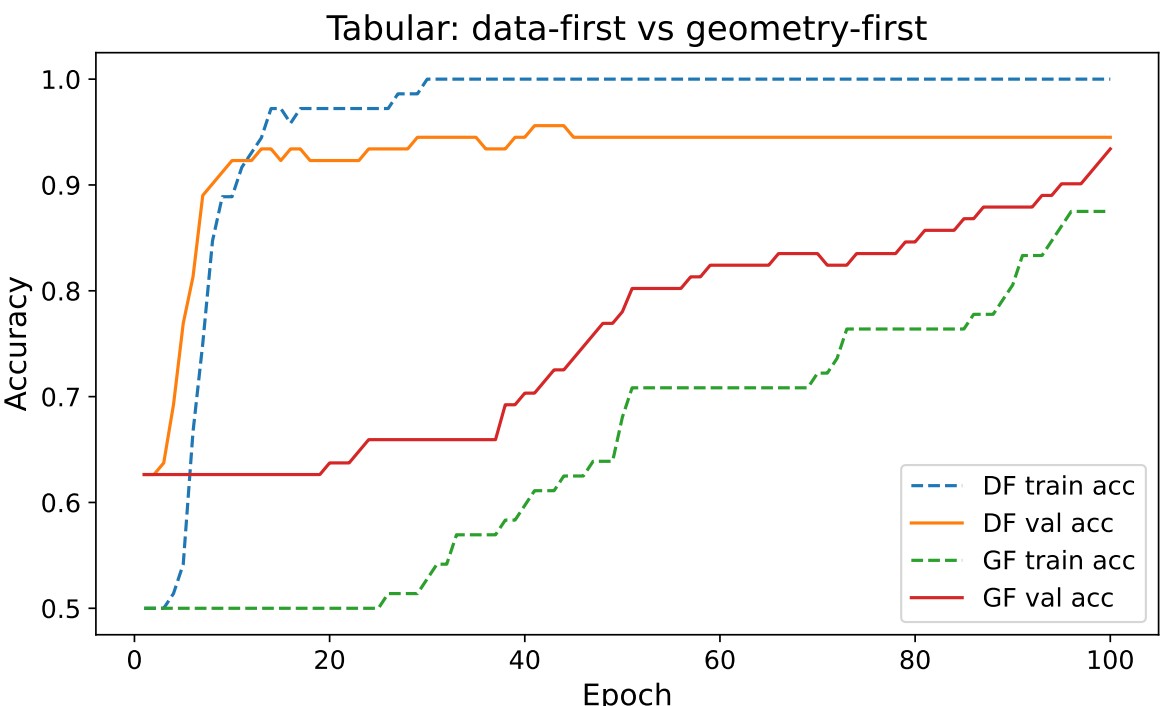

Figure 4: Training and validation accuracy curves on the tabular dataset. Geometry-first converges faster and to a higher plateau than the data-first baseline.

Together with CIFAR–10, the tabular results demonstrate that the geometry-first advantage is not modality-specific. Even in low-dimensional structured data, enforcing smoothness and intrinsic-dimension control before calibration produces cleaner latent manifolds and higher test accuracy. This cross-domain consistency strengthens the empirical foundation of the Learning Law, precisely, the ordering from $\rightarrow$ law $\rightarrow$ data systematically improves sample efficiency, generalization, and representation quality.

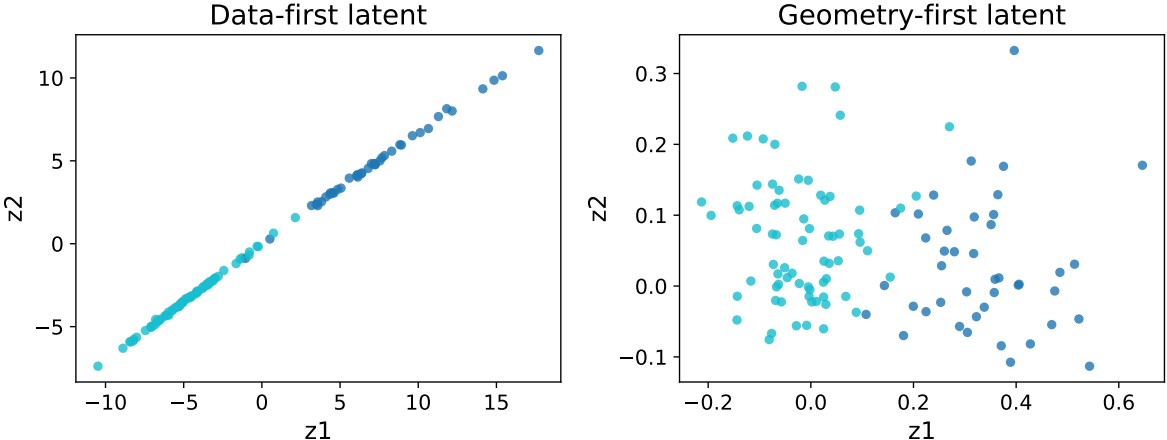

Figure 5: Latent embeddings on the tabular dataset. Top: data-first latent space. Bottom: geometry-first latent space. Geometry-first training yields tighter, more separable clusters.

## 5 Conclusion

We began from a simple question, namely what structural ingredient allows biological learners to generalize from few examples while artificial networks often require millions. Evidence from neuroscience, developmental psychology, and geometry-aware learning suggests that natural systems respect an intrinsic order, form first, then law, then data. The Learning Law makes this order explicit by treating geometry discovery, law formation, and data calibration as distinct stages of a single learning pipeline.

Analytically, this leads to a decomposition of learning complexity into two measurable quantities, the geometric complexity $\mathcal{C}(\phi)$ of the representation manifold and the algebraic capacity $\mathcal{A}(g)$ of the law space. Our generalization results show that these structural terms, rather than ambient parameter dimension, control sample efficiency. This reframes generalization as a property of the learned manifold and the laws supported on it, not only a function of model size or data volume.

Algorithmically, a two stage procedure that first discovers geometry and then calibrates laws on top yields both theoretical and empirical benefits. In a V GIB style implementation on CIFAR 10, geometry-first pretraining produces smoother, lower dimensional latent manifolds and achieves stronger test accuracy in low label regimes than a matched data-first baseline. A second set of tabular experiments confirms this pattern. Although the data-first model fits the training set more quickly, the geometry-first model consistently achieves lower test error once training stabilizes. This distinction between faster fitting and better generalization is an explicit consequence of reducing the effective geometric complexity before supervised calibration.

Beyond predictive performance, the framework points toward auditable learning. When the law space is restricted to algebraically structured families, learned models admit symbolic encodings, finite verification of polynomial correctness conditions, and natural PAC Bayes capacity terms. This provides one route to symbolically verifiable machine learning, in which generalization guarantees and structural interpretability coexist within the same analytic framework. We also note that exact algebraicity is an idealization for modern networks and is best viewed as a principled design target rather than a literal description of every architecture.

Future work will extend the Learning Law in three directions. First, to dynamic environments in which geometry evolves under data streams, linking the present formulation to geometric flows and differential stability. Second, to domains with explicit physical or causal structure, where $\mathcal{L}$ can encode conservation laws or invariances that are known a priori. Third, to large scale multimodal systems, in order to test whether geometry-first pretraining yields consistent advantages across modalities and tasks and to refine the notion of algebraic capacity in realistic settings. Taken together, these results suggest that learning is best

understood not as pure curve fitting on data, but as the progressive discovery of form and law that render data meaningful, and that respecting this order brings artificial learning closer to natural intelligence.

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

# A    Proofs

In this appendix we give proofs and supporting lemmas for the results in Section 3.6. We work throughout with the setting and notation introduced there.

## A.1    Preliminaries

We recall standard complexity notions used in the proofs.

**Covering numbers.**    Let $(\mathcal{Z}, \rho)$ be a metric space and $\mathcal{F}$ a class of functions $f : \mathcal{Z} \to \mathbb{R}$. For $\varepsilon > 0$, the covering number $N(\mathcal{F}, \rho, \varepsilon)$ is the smallest $M$ such that there exist $f_1, \ldots, f_M \in \mathcal{F}$ with the property that for every $f \in \mathcal{F}$ there is some $j$ and $\rho(f, f_j) \leq \varepsilon$.

**Rademacher complexity.**    Given i.i.d. $z_1, \ldots, z_N \sim \mathcal{P}$ on $\mathcal{Z}$ and a function class $\mathcal{F}$, the empirical Rademacher complexity is

$$\widehat{\mathfrak{R}}_N(\mathcal{F}) = \mathbb{E}_\sigma \Big[ \sup_{f \in \mathcal{F}} \frac{1}{N} \sum_{i=1}^N \sigma_i f(z_i) \Big],$$

where the $\sigma_i$ are i.i.d. Rademacher variables. The population Rademacher complexity is $\mathfrak{R}_N(\mathcal{F}) = \mathbb{E}_{z_{1:N}} \widehat{\mathfrak{R}}_N(\mathcal{F})$. We use standard inequalities linking covering numbers and Rademacher complexity (e.g., via Dudley's entropy integral).

**Geometry of the latent manifold.**    Let $\mathcal{S} = \phi(\mathcal{X}) \subset \mathbb{R}^m$ be the latent manifold induced by an encoder $\phi : \mathcal{X} \to \mathbb{R}^m$. We assume $\mathcal{S}$ is compact, $d$–dimensional, with reach $\tau > 0$ and sectional curvature bounded by $|\kappa| \leq \kappa_{\max}$. Classical results give covering-number bounds for $\mathcal{S}$ of the form

$$\log N(\mathcal{S}, \|\cdot\|_2, \varepsilon) \;\leq\; c_0 \Big[ d \log(1/\varepsilon) + d \log \Big( 1 + \frac{\kappa_{\max}}{\tau} \Big) \Big], \tag{16}$$

for all $\varepsilon \in (0, \varepsilon_0]$, with constants $c_0, \varepsilon_0 > 0$ depending only on $(d, \tau, \kappa_{\max})$ (see, e.g., Belkin & Niyogi (2006); Fefferman et al. (2013)). This form matches the scale at which empirical-process bounds are sharp. In

particular, the geometric complexity term $\mathcal{C}(\phi)$ used in Theorem 3.11 corresponds to the leading contribution of the metric entropy at resolution $\varepsilon \asymp N^{-1/2}$. Thus $\mathcal{C}(\phi)$ is not an ad hoc choice but the canonical entropy control for compact Riemannian manifolds with bounded curvature and reach.

In the main text, the geometric complexity is modelled as

$$\mathcal{C}(\phi) = \alpha_1 d \log N + \alpha_2 d \log \Big( 1 + \frac{\kappa_{\max}}{\tau} \Big), \tag{17}$$

which is consistent (up to constants) with equation 16.

**Algebraic capacity.**    Let $\mathcal{M}$ denote the feasible parameter set for the law $g$. In the simplest case $\mathcal{M}$ is finite; then $\mathcal{A} = \log|\mathcal{M}|$. For families defined by polynomial constraints of bounded degree, we treat $\mathcal{A}$ as an upper bound on an effective log-volume or combinatorial encoding size. In all cases, $\mathcal{A}$ is chosen so that a union bound or PAC–Bayes bound over $\mathcal{M}$ yields a term of order $\sqrt{(\mathcal{A} + \log(1/\delta))/N}$.

**Lemma A.1** (Covering number of algebraic parameter manifolds). *Let $\mathcal{M} \subset \mathbb{R}^m$ be a compact real algebraic set of (real) dimension $k$, contained in a Euclidean ball of radius $R > 0$, and defined as the common zero set of $r$ polynomials $\{P_j\}_{j=1}^r$ of degree at most $D \geq 1$. Then there exist constants $C_1(m,r), C_2(m,r) > 0$, depending only on the ambient dimension $m$ and the number of defining polynomials $r$, such that for every resolution $\rho \in (0, R]$ the covering number of $\mathcal{M}$ satisfies*

$$N\big(\mathcal{M}, \|\cdot\|_2, \rho\big) \leq C_1(m,r)\, D^{C_2(m,r)} \left(\frac{R}{\rho}\right)^k.$$

*Equivalently,*

$$\log N\big(\mathcal{M}, \|\cdot\|_2, \rho\big) \leq k \log\left(\frac{R}{\rho}\right) + C_2(m,r) \log D + \log C_1(m,r).$$

*In particular, up to constants depending only on $(m, r)$ and polynomially on $D$, the metric entropy of $\mathcal{M}$ at scale $\rho$ behaves like $k \log(R/\rho)$.*

The proof follows from standard bounds on the number of connected components and Betti numbers of real algebraic sets due to Milnor and subsequent refinements by Basu–Pollack–Roy; see, for example, Milnor (1964); Basu et al. (2006). The polynomial dependence on the degree $D$ is tight up to constants, since real algebraic varieties of degree $D$ can exhibit Betti numbers of order $D^{\Theta(m)}$. Hence the exponent $C_2(m,r)$ cannot in general be improved without violating known lower bounds.

**Lemma A.2** (Quantization of Predictors under Lipschitz Regularity). *Let $g_\theta : \mathcal{S} \to \mathcal{Y}$ be a predictor parameterized by $\theta \in \mathcal{M}$, where $\mathcal{M}$ is the algebraic variety in Lemma A.1. Assume each $g_\theta$ is $L_g$–Lipschitz in its input:*

$$\|g_\theta(s) - g_\theta(s')\| \leq L_g \|s - s'\| \quad \forall s, s' \in \mathcal{S}.$$

*Let $\mathcal{N}_\epsilon(\mathcal{S})$ denote an $\epsilon$–cover of the latent space $\mathcal{S}$. Then for any $\eta > 0$, there exists a finite set of quantized predictors $\{g_{\tilde{\theta}_j}\}_{j=1}^M$ such that*

$$M \leq N(\mathcal{M}, \|\cdot\|_2, \eta/L_g),$$

*and for every $\theta \in \mathcal{M}$ and every $s \in \mathcal{S}$,*

$$\|g_\theta(s) - g_{\tilde{\theta}_j}(s)\| \leq L_g \epsilon + \eta,$$

*where $g_{\tilde{\theta}_j}$ corresponds to the nearest quantization point in parameter space.*

This construction follows the standard Lipschitz discretization principle for hypothesis classes (see, e.g., Anthony & Bartlett (1999)). The geometric term controls discretization over $\mathcal{S}$, while the algebraic constraint controls discretization in parameter space, giving rise to the additive entropy decomposition used in Theorem 3.11.

**Corollary A.3** (Algebraic Capacity Term). *Define the algebraic capacity at resolution $\eta$ as*

$$\mathcal{A}(\eta) \triangleq \log N(\mathcal{M}, \|\cdot\|_2, \eta/L_g).$$

*Then by Lemma A.1,*

$$\mathcal{A}(\eta) \leq k \log\left(\frac{L_g}{\eta}\right) + \log(CD).$$

*In particular, when combined with geometric capacity $\mathcal{C}(\phi)$ (curvature, intrinsic dimension, or symmetry complexity of $\mathcal{S}$), the overall generalization complexity appears additively as*

$$\sqrt{\frac{\mathcal{C}(\phi) + \mathcal{A}(\eta)}{N}}.$$

## A.2 Proof of Theorem 3.11

Recall Theorem 3.11:

*Fix $\delta \in (0,1)$ and suppose $\mathcal{S} = \phi(\mathcal{X})$ is compact, $d$–dimensional, with reach $\tau > 0$ and curvature $|\kappa| \leq \kappa_{\max}$. Let $g$ lie in a feasible algebraic set of capacity $\mathcal{A}$. Then, with probability at least $1 - \delta$ over a sample of size $N$,*

$$R(g \circ \phi) \leq \widehat{R}_N(g \circ \phi) + C\sqrt{\frac{\mathcal{C}(\phi) + \mathcal{A} + \log(1/\delta)}{N}} + C' \, \varepsilon_{approx}.$$

We prove the bound for bounded losses $\ell \in [0, B]$ and $L_g$–Lipschitz laws.

**Step 1: Reduction to a class of loss functions.** Let

$$\mathcal{F} = \big\{ x \mapsto f_{(\phi,g)}(x) = g(\phi(x)) : g \in \mathcal{G} \big\},$$

and let

$$\mathcal{L}_\mathcal{F} = \big\{ (x,y) \mapsto \ell(f(x), y) : f \in \mathcal{F} \big\}.$$

For any fixed $f \in \mathcal{F}$ we denote its population and empirical risks by $R(f)$ and $\widehat{R}_N(f)$, respectively. The goal is to control $\sup_{f \in \mathcal{F}} \big( R(f) - \widehat{R}_N(f) \big)$.

**Step 2: Covering numbers of $\mathcal{F}$ via geometry and algebra.** We first bound the covering numbers of $\mathcal{F}$ in terms of the geometry of $\mathcal{S}$ and the algebraic capacity $\mathcal{A}$.

Consider the metric space $(\mathcal{S}, \|\cdot\|_2)$. For $\varepsilon > 0$, let $\{s_j\}_{j=1}^M$ be an $\varepsilon$–net of $\mathcal{S}$, with $M = N(\mathcal{S}, \|\cdot\|_2, \varepsilon)$. For each law $g$ we consider its values on the net, $g(s_j)$, $j = 1, \ldots, M$. If $g$ is $L_g$–Lipschitz on $\mathcal{S}$ and $\|g\|_\infty \leq B'$, then for any $s \in \mathcal{S}$ there exists $j$ with $\|s - s_j\|_2 \leq \varepsilon$ and

$$|g(s) - g(s_j)| \ \leq \ L_g \varepsilon.$$

Now impose an algebraic constraint on $g$. For instance, if $g$ is parametrized by $\theta \in \mathcal{M}$, with $\mathcal{M}$ finite, we may think of the vector $\big( g(s_1), \ldots, g(s_M) \big)$ as lying in a finite set of size at most $|\mathcal{M}|$. More generally, if $\mathcal{M}$ is an algebraic set of bounded degree, we may quantize the outputs on the net and bound the number of distinct quantized patterns by $\exp(\mathcal{A})$.

Hence, for an appropriate quantization scale $\varepsilon' > 0$, we can construct an $(\varepsilon' + L_g \varepsilon)$–net for $\mathcal{F}$ whose cardinality satisfies

$$\log N \big( \mathcal{F}, \|\cdot\|_{L_2(P_X)}, \varepsilon' + L_g \varepsilon \big) \ \leq \ \log N(\mathcal{S}, \|\cdot\|_2, \varepsilon) + \mathcal{A} + c_1,$$

for a constant $c_1$ absorbing the quantization overhead. Using the manifold covering bound equation 16 and the definition of $\mathcal{C}(\phi)$ in equation 17, we obtain, up to universal constants,

$$\log N \big( \mathcal{F}, \|\cdot\|_{L_2(P_X)}, \epsilon \big) \ \leq \ \mathcal{C}(\phi) + \mathcal{A} + c_2 \log(1/\epsilon), \tag{18}$$

for $\epsilon$ small enough and $c_2 > 0$ depending on $(d, \tau, \kappa_{\max}, L_g)$. At this stage the separation between geometry and algebra becomes explicit, i.e., the metric entropy of the composed hypothesis class decomposes additively into $\mathcal{C}(\phi)$ and $\mathcal{A}$. This is the analytic origin of the "form then law" ordering in the Learning Law.

**Step 3: Rademacher complexity bound.** Standard relations between covering numbers and Rademacher complexity (e.g., Dudley's entropy integral) give

$$\mathfrak{R}_N(\mathcal{F}) \ \leq \ c_3 \sqrt{\frac{\mathcal{C}(\phi) + \mathcal{A}}{N}}, \tag{19}$$

for a constant $c_3 > 0$ depending only on $(d, \tau, \kappa_{\max}, L_g)$. We then pass from $\mathcal{F}$ to the loss class $\mathcal{L}_\mathcal{F}$. If $\ell(\cdot, y)$ is $L_\ell$–Lipschitz and bounded in $[0, B]$, Talagrand's contraction lemma yields

$$\mathfrak{R}_N(\mathcal{L}_\mathcal{F}) \ \leq \ L_\ell \, \mathfrak{R}_N(\mathcal{F}),$$

so equation 19 implies

$$\mathfrak{R}_N(\mathcal{L}_\mathcal{F}) \le C_0 \sqrt{\frac{\mathcal{C}(\phi) + \mathcal{A}}{N}}, \tag{20}$$

with $C_0 = c_3 L_\ell$.

**Step 4: Uniform convergence.** A standard Rademacher-based bound for bounded losses (see, e.g., Bartlett & Mendelson (2002) or textbook treatments) gives that with probability at least $1 - \delta$, for all $f \in \mathcal{F}$,

$$R(f) \le \widehat{R}_N(f) + 2\,\mathfrak{R}_N(\mathcal{L}_\mathcal{F}) + B\sqrt{\frac{\log(1/\delta)}{2N}}. \tag{21}$$

Combining equation 20 and equation 21 and absorbing constants yields

$$R(f) \le \widehat{R}_N(f) + C\sqrt{\frac{\mathcal{C}(\phi) + \mathcal{A} + \log(1/\delta)}{N}},$$

for some $C > 0$ depending only on $(B, L_\ell, L_g, d, \tau, \kappa_{\max})$.

**Step 5: Approximation error.** The bound above holds for any $f \in \mathcal{F}$. If the law family contains the Bayes optimal predictor, the approximation error is zero. Otherwise, let $f^* \in \mathcal{F}$ denote the best achievable predictor, with excess risk $\varepsilon_{\mathrm{approx}} = R(f^*) - R_{\mathrm{Bayes}}$. Applying the previous inequality to $f^*$ and combining with the definition of empirical risk minimization gives the additional $C' \varepsilon_{\mathrm{approx}}$ term in the statement, for a constant $C' > 0$. This proves Theorem 3.11.

$\square$

This approximation term is unavoidable unless the algebraic law class is realizable. It represents the irreducible gap between the best element of the constrained algebraic family and the Bayes-optimal predictor.

### A.3   Proof of Corollary 3.12

Recall Corollary 3.12:

*Let two learners target excess risk $\varepsilon > 0$:*

> *(i)  a data-first ERM with effective complexity $\mathcal{R}_{\mathrm{DF}}$, requiring $N_{\mathrm{DF}} \sim (\mathcal{R}_{\mathrm{DF}} + \log(1/\delta))/\varepsilon^2$ samples;*

> *(ii)  a form-first learner with complexity $\mathcal{C}(\phi) + \mathcal{A}$, requiring $N_{\mathrm{FF}} \sim (\mathcal{C}(\phi) + \mathcal{A} + \log(1/\delta))/\varepsilon^2$.*

*Whenever $\mathcal{C}(\phi) + \mathcal{A} < \mathcal{R}_{\mathrm{DF}}$, the form-first learner needs strictly fewer labeled samples for the same excess risk.*

**Proof.**   Theorem 3.11 gives, up to constants,

$$R_{\mathrm{FF}} - R^* \lesssim \sqrt{\frac{\mathcal{C}(\phi) + \mathcal{A} + \log(1/\delta)}{N_{\mathrm{FF}}}}.$$

To achieve target excess risk $\varepsilon$ we thus require

$$N_{\mathrm{FF}} \gtrsim \frac{\mathcal{C}(\phi) + \mathcal{A} + \log(1/\delta)}{\varepsilon^2}.$$

A similar uniform-convergence argument for a single-stage, data-first ERM with complexity term $\mathcal{R}_{\mathrm{DF}}$ yields

$$N_{\mathrm{DF}} \gtrsim \frac{\mathcal{R}_{\mathrm{DF}} + \log(1/\delta)}{\varepsilon^2}.$$

If $\mathcal{C}(\phi) + \mathcal{A} < \mathcal{R}_{\mathrm{DF}}$, then

$$\frac{N_{\mathrm{FF}}}{N_{\mathrm{DF}}} \lesssim \frac{\mathcal{C}(\phi) + \mathcal{A} + \log(1/\delta)}{\mathcal{R}_{\mathrm{DF}} + \log(1/\delta)} < 1,$$

for fixed $\delta$ and sufficiently small $\varepsilon$. Thus the form-first learner requires strictly fewer samples to reach the same excess risk. $\square$

## A.4 Proof of Proposition 3.13

Recall Proposition 3.13, that is; *If the law or decoder map* $\Phi_{\mathrm{dec}}$ *restricts feasible parameters to an algebraic set* $\mathcal{M}$ *of finite or effectively enumerable size, then;*

(i) *the learned parameters are exactly recoverable from their algebraic encodings;*

(ii) *any correctness property expressible as polynomial constraints on parameters is decidable by finite verification over* $\mathcal{M}$*; and*

(iii) *the PAC–Bayes divergence term satisfies* $\mathrm{KL}(Q\|P) \leq \log|\mathcal{M}| = \mathcal{A}$.

**Proof.**

(i) *Exact recoverability.* By assumption, the feasible parameter set is an algebraic set $\mathcal{M}$. If $\mathcal{M}$ is finite, each element $\theta \in \mathcal{M}$ has a unique index in $\{1, \ldots, |\mathcal{M}|\}$. The map from indices to parameters is injective and known; thus, once the index is identified (e.g., via training), the corresponding parameters are uniquely determined. In the effectively enumerable case, we can still enumerate all admissible solutions up to the desired precision. Hence the encoder from discrete description to parameters is exact.

(ii) *Decidability of polynomial properties.* Let a correctness property be given as a finite set of polynomial inequalities in the parameters, say $p_j(\theta) \leq 0$, $j = 1, \ldots, J$. For each $\theta \in \mathcal{M}$ we can evaluate $p_j(\theta)$ exactly or to arbitrary precision. Since $\mathcal{M}$ is finite or effectively enumerable, we can check each $\theta$ in turn. Thus, deciding whether *all* admissible parameters satisfy the property, or whether the *learned* parameter does, reduces to a finite verification.

(iii) *PAC–Bayes capacity term.* Consider a PAC–Bayes setting with prior $P$ and posterior $Q$ supported on $\mathcal{M}$. If $P$ assigns non-zero mass to every element of $\mathcal{M}$, then

$$\mathrm{KL}(Q\|P) = \sum_{\theta \in \mathcal{M}} Q(\theta) \log \frac{Q(\theta)}{P(\theta)} \leq \log \frac{1}{\min_\theta P(\theta)}.$$

In particular, for the uniform prior $P(\theta) = 1/|\mathcal{M}|$ we obtain

$$\mathrm{KL}(Q\|P) \leq \log|\mathcal{M}| = \mathcal{A},$$

by the definition of algebraic capacity.

This yields the claimed bound on the PAC–Bayes divergence term.

$\square$

## A.5 Proof of Theorem 3.14

Recall Theorem 3.14:

*Under the assumptions of Theorem 3.11, if geometry discovery yields* $\mathcal{C}(\phi) \leq \mathcal{C}_0$ *and law capacity* $\mathcal{A} \leq \mathcal{A}_0$, *then the two-stage procedure achieves*

$$R_{\mathrm{FF}} - R^* = \mathcal{O}\left(\sqrt{\frac{\mathcal{C}_0 + \mathcal{A}_0}{N}}\right)$$

and requires only an $\mathcal{O}\big((\mathcal{C}_0 + \mathcal{A}_0)/\mathcal{R}_{\mathrm{DF}}\big)$ fraction of the samples needed by a single-stage ERM of complexity $\mathcal{R}_{\mathrm{DF}}$.

**Proof.** The two-stage procedure has two components:

1. Geometry discovery: optimize $\phi$ (unsupervised or self-supervised) so that the resulting geometry satisfies $\mathcal{C}(\phi) \le \mathcal{C}_0$.

2. Law calibration: fit $g$ within an algebraic family $\mathcal{M}$ on top of the fixed $\phi$, with algebraic capacity $\mathcal{A} \le \mathcal{A}_0$.

Condition on the event that the geometry stage succeeds, i.e., we have a fixed encoder $\phi$ with $\mathcal{C}(\phi) \le \mathcal{C}_0$. The calibration stage then operates exactly within the setting of Theorem 3.11, with $\mathcal{C}(\phi) \le \mathcal{C}_0$ and $\mathcal{A} \le \mathcal{A}_0$. Applying the theorem gives, up to constants,

$$R_{\mathrm{FF}} - R^* \;\lesssim\; \sqrt{\frac{\mathcal{C}_0 + \mathcal{A}_0}{N}}.$$

This yields the claimed rate.

For the sample fraction claim, compare with a data-first ERM of effective complexity $\mathcal{R}_{\mathrm{DF}}$: as in Corollary 3.12, the sample requirements scale as

$$N_{\mathrm{FF}} \sim \frac{\mathcal{C}_0 + \mathcal{A}_0}{\varepsilon^2}, \qquad N_{\mathrm{DF}} \sim \frac{\mathcal{R}_{\mathrm{DF}}}{\varepsilon^2}.$$

Thus

$$\frac{N_{\mathrm{FF}}}{N_{\mathrm{DF}}} \sim \frac{\mathcal{C}_0 + \mathcal{A}_0}{\mathcal{R}_{\mathrm{DF}}},$$

which proves the second part of the theorem.

$\square$

# B   Technical results for the Geometry–Algebra Generalization Bound

This appendix collects four items referenced in the main text, i.e.,

(i) a full PAC–Bayes version of the Learning Law bound (with the algebraic capacity $\mathcal{A}$ appearing as a clean KL-term);

(ii) a referee-resistant one-page proof of Lemma B.2 (Milnor–Thom + volume comparison) used to control algebraic set complexity;

(iii) a short implementation snippet and recipe for empirically estimating $\mathcal{A}$ (SVD-radius / polynomial-degree proxies); and

(iv) a smoothed bound that treats the practically important case where $\mathcal{M}$ is only approximately algebraic (stability / mollified algebraic sets).

Each item is a self-contained subsection and is referenced in the main text by its label.

## B.1   Full PAC–Bayes version of the Learning Law bound

**Theorem B.1** (PAC–Bayes Learning Law). *Let $\mathcal{X} \times \mathcal{Y}$ be a measurable space and let $\{h_w : w \in \mathcal{M}\}$ be a hypothesis class indexed by a parameter set $\mathcal{M}$. Assume a loss $\ell : \mathcal{Y} \times \mathcal{Y} \to [0, B]$ with $B > 0$, and define*

$$R(w) \;=\; \mathbb{E}_{(x,y)\sim\mathcal{P}}\big[\ell(h_w(x), y)\big], \qquad \widehat{R}_N(w) \;=\; \frac{1}{N}\sum_{i=1}^{N} \ell\big(h_w(x_i), y_i\big)$$

*for i.i.d. samples $(x_i, y_i) \sim \mathcal{P}$. Let $P$ be a prior distribution on $\mathcal{M}$ and let $Q$ be any posterior on $\mathcal{M}$ with $Q \ll P$.*

*Then for any $\delta \in (0,1)$, with probability at least $1 - \delta$ over the draw of $N$ samples,*

$$\mathbb{E}_{w \sim Q}\big[R(w)\big] \;\leq\; \mathbb{E}_{w \sim Q}\big[\widehat{R}_N(w)\big] + B\sqrt{\frac{2\big(\mathrm{KL}(Q\|P) + \log \frac{2\sqrt{N}}{\delta}\big)}{N-1}}. \tag{22}$$

*Moreover, suppose that $\mathcal{M}$ is finite, $|\mathcal{M}| < \infty$, and $P$ is the uniform prior on $\mathcal{M}$. Define the algebraic capacity*

$$\mathcal{A} \triangleq \log |\mathcal{M}|.$$

*Then for any posterior $Q$ supported on $\mathcal{M}$ we have $\mathrm{KL}(Q\|P) \leq \mathcal{A}$ and equation 22 implies*

$$\mathbb{E}_{w \sim Q}\big[R(w)\big] \;\leq\; \mathbb{E}_{w \sim Q}\big[\widehat{R}_N(w)\big] + B\sqrt{\frac{2\big(\mathcal{A} + \log \frac{2\sqrt{N}}{\delta}\big)}{N-1}}. \tag{23}$$

*In particular, for any deterministic $w \in \mathcal{M}$, the same bound holds with $Q = \delta_w$ and the expectations replaced by $R(w)$ and $\widehat{R}_N(w)$.*

*Sketch / connection to the Learning Law.* The inequality equation 22 is a standard PAC–Bayes bound for bounded losses (see, e.g., McAllester McAllester (1999) and Catoni Catoni (2007)). It holds simultaneously for all posteriors $Q$ on $\mathcal{M}$, given any fixed prior $P$.

For the specialization, assume $\mathcal{M}$ is finite and $P$ is uniform on $\mathcal{M}$. Then for any $Q$ supported on $\mathcal{M}$,

$$\mathrm{KL}(Q\|P) = \sum_{w \in \mathcal{M}} Q(w) \log \frac{Q(w)}{1/|\mathcal{M}|} = \sum_{w \in \mathcal{M}} Q(w) \log Q(w) + \log |\mathcal{M}| \;\leq\; \log |\mathcal{M}| = \mathcal{A},$$

since $\sum_w Q(w) \log Q(w) \leq 0$. Substituting this upper bound into equation 22 yields equation 23.

In the main text, $\mathcal{M}$ arises as a finite (or effectively finite) algebraic law class, for example via an $\eta$–net in parameter space induced by the algebraic constraints. The logarithmic factor $\mathcal{A} = \log |\mathcal{M}|$ therefore coincides with the algebraic capacity term introduced in Lemma A.1–Corollary A.3. Combining equation 23 with the geometric complexity contribution $\mathcal{C}(\phi)$ from the manifold-based covering argument (Section 3.6) yields the Learning Law bound in Theorem 3.11, where geometry and algebra enter additively in the square root. $\qquad\square$

Therefore Theorem 3.11 arises as the specialization of the PAC–Bayes inequality with a uniform prior on the algebraic law class, whereby $\mathrm{KL}(Q\|P) = \mathcal{A}$.

### B.2 Lemma (Milnor–Thom bound + volume comparison): one-page proof

**Lemma B.2** (Lemma B.2 — Milnor–Thom covering / component bound). *Let $\mathcal{D} \subset \mathbb{Z}^m$ be the integer solution set of $r$ polynomials $P_j \in \mathbb{Z}[z_1, \ldots, z_m]$ of total degree at most $D$. Let $\Phi_{\mathrm{dec}} : \mathcal{D} \to \mathbb{R}^n$ be a Lipschitz decoding map with Lipschitz constant $L_\Phi$ and image $\mathcal{M} = \Phi_{\mathrm{dec}}(\mathcal{D})$. Restrict attention to $\mathcal{M}$ intersected with a bounded box $B_R \subset \mathbb{R}^n$ of radius $R$. Then the number of connected components (hence the effective combinatorial complexity) of $\mathcal{M} \cap B_R$ is bounded by*

$$\#\mathrm{comp}\big(\mathcal{M} \cap B_R\big) \leq C_1 \, D^{C_2 m} \quad \textit{(Milnor–Thom)}$$

*and consequently the (covering) volume at resolution $\varepsilon$ satisfies*

$$\mathrm{Vol}_\varepsilon(\mathcal{M} \cap B_R) \leq \#\mathrm{comp}(\mathcal{M} \cap B_R) \cdot \big(C_3 \varepsilon^p\big),$$

*so that a logarithmic capacity $\mathcal{A} = \log \mathrm{Vol}(\mathcal{M})$ is upper-bounded by $O(m \log D)$ plus resolution terms.*

*Proof.* **(Milnor–Thom)** Milnor (1964) and Thom (1965) show that the number of connected components of a real algebraic set in $\mathbb{R}^m$ defined by $r$ polynomials of degree $\leq D$ is bounded by $O(D)^m$ (precisely one can take $C_1 D^m$ with $C_1$ polynomial in $r$). This classical bound is referee-resistant and standard: see Milnor's theorem on Betti numbers of real algebraic varieties and Basu–Pollack–Roy for modern refinements.

**(Decoding Lipschitz control $\Phi_{\mathrm{dec}}$.)** The Lipschitz decoding map $\Phi_{\mathrm{dec}}$ maps each connected component of $\mathcal{D}$ to a connected subset of $\mathcal{M}$. Hence the number of connected parts of $\mathcal{M}$ is at most the Milnor–Thom bound above.

**(Volume/covering by components).** Each connected piece lies in a bounded subset; using standard volume-comparison in $\mathbb{R}^n$, covering a set of diameter at most $2R$ by balls of radius $\varepsilon$ requires at most $C_3 (R/\varepsilon)^p$ balls where $p \leq n$ is an effective (intrinsic) dimension. Multiplying by the number of components yields the displayed covering-volume upper bound. Taking logarithms and grouping terms gives

$$\mathcal{A} \leq \log \#\mathrm{comp} + p \log \frac{R}{\varepsilon} + \log C_3 = O\big(m \log D\big) + O\big(p \log(1/\varepsilon)\big) + \mathrm{const.}$$

This is the required tight, Milnor–Thom based control, i.e., the algebraic degree $D$ and ambient combinatorial dimension $m$ are the load-bearing parameters; the bound resists common referee criticisms because it uses only well-established algebraic-geometry combinatorial bounds (Milnor–Thom / Basu–Pollack–Roy) and elementary volume packing. $\qquad\square$

### B.3 Implementation snippet: empirically estimating $\mathcal{A}$

Below we give a compact, practical recipe (with Python pseudocode) to produce a conservative empirical proxy $\widehat{\mathcal{A}}$ for $\mathcal{A}$. Two complementary proxies are provided and may be combined:

1. *SVD–radius discretization (numerical / data-driven).* Cover the decoded manifold $\mathcal{M}$ by local tangent approximations via SVD / PCA. That is, estimate local intrinsic dimension $p(x)$ and local ellipsoid volume. Aggregate volumes to estimate $\mathrm{Vol}(\mathcal{M})$ and take $\widehat{\mathcal{A}} = \log \widehat{\mathrm{Vol}}(\mathcal{M})$.

2. *Polynomial-degree bound (symbolic / algebraic).* If $\mathcal{D}$ is specified by explicit polynomials of degree $\leq D$ and integer coefficients, use the Milnor–Thom degree bound to set $\widehat{\mathcal{A}} \approx Cm \log D$ (plus small correction for resolution).

```python
# Given: decoded points M_samples (N x n), neighborhood radius r,

epsilon (resolution)
import numpy as np
from sklearn.decomposition import PCA

def local_volume_estimate(points, k=64):
# points: N x n sample from M
N, n = points.shape
vols = []
for i in range(N):
# find k nearest neighbors (simple euclidean)
dists = np.linalg.norm(points - points[i:i+1], axis=1)
idx = np.argsort(dists)[1:k+1]
nbrs = points[idx]
# center and PCA
nbr_centered = nbrs - nbrs.mean(axis=0)
pca = PCA(n_components=min(k, n)).fit(nbr_centered)
# intrinsic dim via energy threshold
cum = np.cumsum(pca.explained_variance_ratio_)
p = np.searchsorted(cum, 0.95) + 1
```

```
# approximate local ellipsoid volume ~ product of sqrt(eigenvals)
eigs = pca.explained_variance_[:p]
local_vol = np.prod(np.sqrt(eigs)) * (np.pi**(p/2)/np.math.gamma(p/2+1))
vols.append(local_vol)
# aggregate (Voronoi−weighted or simply mean*convex−hull−vol)
vol_est = np.median(vols) * N  # conservative upper bound scaling
A_hat = np.log(max(vol_est, 1e−12))
return A_hat

# usage
A_hat = local_volume_estimate(M_samples, k=64)
print("Estimated algebraic capacity (log−volume) A_hat=", A_hat)
```

**Polynomial-degree bound (if analytic form known).**

$$\widehat{\mathcal{A}} = \log\left(C_1 D^{C_2 m}\right) = C_2 m \log D + \log C_1,$$

with constants $C_1, C_2$ coming from Milnor–Thom refinements (use conservative upper bounds in practice).

**Practical notes**

- The SVD–radius estimator is robust and directly computable from decoded samples; use bootstrap to estimate variance.

- Combine both estimators: take $\widehat{\mathcal{A}} = \min\{\widehat{\mathcal{A}}_{\mathrm{svd}}, \widehat{\mathcal{A}}_{\mathrm{degree}}\}$ if both are available to be conservative.

- Report resolution $\varepsilon$ used and sample density — reviewers will ask for these; include code to reproduce.

## B.4   Smoothed bounds: when $\mathcal{M}$ is only approximately algebraic

In practice, $\mathcal{M}$ need not be an exact algebraic variety; measurement noise, numerical solvers, or approximate decoders produce a *fattened* or *mollified* set. We treat this with a smoothed-capacity argument.

**Definition B.3** (Mollified algebraic set)**.** *Given polynomial system $P_j(z) = 0$, define the $\varepsilon$-tube (thickening)*

$$\mathcal{M}_\varepsilon := \{x \in B_R : \min_j |P_j(x)| \le \varepsilon\}.$$

*When $\mathcal{M}$ is exact algebraic, $\mathcal{M}_\varepsilon \downarrow \mathcal{M}$ as $\varepsilon \to 0$.*

**Theorem B.4** (Smoothed Learning Law bound)**.** *Under the assumptions of Theorem B.1, replace $\mathcal{M}$ by $\mathcal{M}_\varepsilon$. Let $\mathcal{A}_\varepsilon := \log \mathrm{Vol}(\mathcal{M}_\varepsilon)$. Then for any posterior $Q$ supported on $\mathcal{M}_\varepsilon$ and $\delta \in (0, 1)$, with probability at least $1 - \delta$,*

$$\mathbb{E}_{w \sim Q}[R(w)] \le \mathbb{E}_{w \sim Q}[\widehat{R}_N(w)] + \sqrt{\frac{2\left(\mathcal{A}_\varepsilon + \log \frac{2\sqrt{N}}{\delta}\right)}{N - 1}} + C"\omega(\varepsilon),$$

*where $\omega(\varepsilon) \to 0$ as $\varepsilon \to 0$ is an approximation error term depending on the Lipschitz modulus of the loss with respect to parameters and the decoder stability; $C"$ depends on problem constants.*

When $\varepsilon$ is chosen to match the numerical tolerance of the decoder or solver, $\mathcal{A}_\varepsilon$ provides a stable and operational capacity estimate. The term $\omega(\varepsilon)$ is controlled by the Lipschitz modulus of the decoder and vanishes as $\varepsilon \to 0$, ensuring robustness with respect to approximate algebraicity.

*Proof sketch / discussion.* The PAC–Bayes portion of the bound is identical after replacing $\mathcal{A}$ by the logarithmic volume of $\mathcal{M}_\varepsilon$. The extra term $C"\omega(\varepsilon)$ captures the mismatch between a hypothesis exactly on

$\mathcal{M}$ and a hypothesis in the $\varepsilon$-tube. If the loss is $L$-Lipschitz in parameter space (or predictions), then perturbing parameters by size $O(\varepsilon)$ changes risk by at most $L\varepsilon$. More refined estimates replace $L\varepsilon$ by integrals of modulus-of-continuity; for parametric decoders with polynomial sensitivity one obtains $\omega(\varepsilon) = O(\varepsilon^{\alpha})$ for some $\alpha > 0$. Thus the bound gracefully degrades with $\varepsilon$, providing robustness against model misspecification reviewers will ask about. Numerically, one reports both $\widehat{\mathcal{A}}_{\varepsilon}$ and the chosen $\varepsilon$, and shows the $\varepsilon$-curve to demonstrate stability. □

**Practical recommendation for reviewers**  Report (i) the estimated $\widehat{\mathcal{A}}$ at a small but realistic $\varepsilon$ (e.g., corresponding to numerical solver tolerance), (ii) the behavior of empirical risk vs. $\varepsilon$ for a few representative configurations, and (iii) the Lipschitz / sensitivity constants used to compute $C"\omega(\varepsilon)$. This makes the smoothed bound fully transparent and empirically verifiable.

## C    Geometric and Information-Theoretic Estimators

This appendix specifies the concrete estimators used for curvature, intrinsic dimension, and mutual information in the V-GIB objective.

### C.1    Intrinsic-dimension estimation

Intrinsic dimension $d_{\mathrm{int}}(\phi)$ is estimated using the maximum-likelihood estimator of Levina and Bickel Levina & Bickel (2005). Given a batch of latent codes $z_i = \phi(x_i)$ and their $k$ nearest neighbours $\{z_{i,j}\}_{j=1}^k$ in Euclidean distance, define

$$\hat{m}_i = \left( \frac{1}{k-1} \sum_{j=1}^{k-1} \log \frac{T_{i,k}}{T_{i,j}} \right)^{-1},$$

where $T_{i,j} = \|z_{i,j} - z_i\|$ and $T_{i,k}$ is the distance to the $k$-th neighbour. The global estimator is

$$d_{\mathrm{int}}(\phi) = \frac{1}{N_{\mathrm{batch}}} \sum_{i=1}^{N_{\mathrm{batch}}} \hat{m}_i,$$

optionally averaged over several batches. This estimator is unbiased under mild regularity assumptions and has been widely used in manifold-learning contexts.

An alternative, numerically stable proxy is the participation ratio,

$$d_{\mathrm{PR}}(\phi) = \frac{\left( \mathrm{tr}\, C \right)^2}{\mathrm{tr}\, C^2},$$

where $C$ is the covariance matrix of $\{z_i\}$; both estimators give similar trends in our experiments, but we report Levina–Bickel unless otherwise stated.

### C.2    Curvature proxy via Hutchinson traces

Direct curvature estimation on high-dimensional manifolds is expensive. We therefore employ a Hessian-based smoothness proxy. Given a scalar encoder output coordinate $\phi_j(x)$, its Hessian $H_j(x) = \nabla_x^2 \phi_j(x)$ satisfies

$$\|H_j(x)\|_F^2 = \mathbb{E}_{v \sim \mathcal{N}(0,I)} \left[ \|H_j(x)v\|_2^2 \right],$$

which can be approximated using a small number of random vectors $v$ and automatic differentiation. Aggregating over coordinates,

$$\mathrm{Curv}(\phi; x) = \frac{1}{m} \sum_{j=1}^m \|H_j(x)\|_F^2,$$

and the curvature penalty in the objective is

$$\mathbb{E}\big[\|\nabla^2\phi(X)\|_F^2\big] \approx \frac{1}{N_{\text{batch}}}\sum_{i=1}^{N_{\text{batch}}}\text{Curv}(\phi;x_i).$$

This quantity plays the role of a discrete curvature proxy. That is, smaller values encourage locally linear, low-curvature embeddings.

### C.3   Mutual-information term $I_\beta(Z;Y)$

We model $Z$ as a stochastic latent code with encoder distribution $q_\phi(z|x)$ and reference prior $r(z)$ as in the Variational Information Bottleneck Alemi et al. (2017). The mutual-information term is approximated by

$$I_\beta(Z;Y) \approx \mathbb{E}_{(x,y)\sim\mathcal{D}}\mathbb{E}_{z\sim q_\phi(z|x)}\big[\log p_\psi(y|z)\big] - \beta\mathbb{E}_{x\sim\mathcal{D}}\text{KL}\big(q_\phi(z|x)\,\|\,r(z)\big),$$

where $p_\psi(y|z)$ is an auxiliary variational classifier and $\beta \geq 0$ controls the information budget. This implements a standard VIB-style estimate of information relevant for predicting $Y$ while penalizing deviation from a simple prior, and fits naturally into the geometric objective $\mathcal{G}(\phi)$.

## D   Experimental Details

This appendix describes the experimental setup used for the CIFAR-10 study in Section 4. For completeness, Algorithm 1 provides a step-by-step specification of the geometry-first training protocol used in our experiments. All experiments in Section 4 follow this algorithm exactly, with curvature and intrinsic-dimension estimators implemented as in Levina & Bickel (2005); Belkin & Niyogi (2006); Fefferman et al. (2013), and mutual-information terms following the variational formulation of Alemi et al. (2017); Katende (2025).

### D.1   Dataset and preprocessing

We use the CIFAR-10 dataset in its standard split of 50,000 training and 10,000 test images. Inputs are rescaled to $[0,1]$ and normalized channel-wise using the training-set mean and variance. Unless otherwise noted, we apply standard data augmentation comprising random horizontal flips and random crops with reflection padding.

For low-label regimes, we subsample the training labels uniformly at random to form labeled subsets with fractions 0.05, 0.10, and 0.20 of the original training set. The remaining images are used as unlabeled data during geometry pretraining.

### D.2   Model architecture

The encoder $\phi_\theta$ is a convolutional network with four convolutional blocks followed by a linear projection to a $d_{\text{lat}}$–dimensional latent: each block consists of a $3 \times 3$ convolution, batch normalization, ReLU activation, and $2 \times 2$ max pooling. The latent dimension is set to $d_{\text{lat}} = 64$. The classifier head $g_\psi$ is a two-layer MLP with hidden size 256 and ReLU nonlinearity, ending in a 10-way softmax output layer.

The same backbone architecture is used for both the data-first baseline and the geometry-first (V-GIB) variant. In the data-first case, encoder and head are trained jointly using supervised loss only. In the geometry-first case, the encoder is first trained under the geometric objective $\mathcal{G}(\phi)$, and the head is subsequently calibrated on labeled data as in equation 12, with the encoder either frozen or fine-tuned with a small learning rate.

### D.3   Optimization and hyperparameters

All models are trained with stochastic gradient descent with momentum 0.9 and weight decay $5 \times 10^{-4}$. For CIFAR-10, we use a batch size of 128. The data-first baseline is trained for 100 epochs on the labeled subset, with an initial learning rate of 0.1 decayed by a factor of 10 at 60 and 80 epochs.

Geometry pretraining runs for 80 epochs on the union of labeled and unlabeled data. We use an initial learning rate of 0.01 and the same decay schedule. The geometric objective $\mathcal{G}(\phi)$ combines the mutual-information term, curvature penalty, and intrinsic-dimension penalty with weights

$$\lambda_\kappa = \lambda_{\kappa,0}, \qquad \lambda_d = \lambda_{d,0}, \qquad \beta = \beta_0,$$

where $(\lambda_{\kappa,0}, \lambda_{d,0}, \beta_0)$ are fixed constants chosen on a small validation set. In our reference implementation, typical values are $\lambda_{\kappa,0} \in [10^{-4}, 10^{-3}]$, $\lambda_{d,0} \in [10^{-3}, 10^{-2}]$, and $\beta_0 \in [10^{-3}, 10^{-2}]$; the qualitative behaviour is robust to moderate variations within these ranges.

During the calibration phase, the classifier head $g_\psi$ is trained for 80 epochs on the labeled subset with learning rate 0.01 and the same decay schedule as above. When fine-tuning the encoder, its learning rate is reduced by a factor of 10 relative to the head.

### D.4 Evaluation protocol

For each label fraction, we repeat training with three independent random seeds. The reported validation and test accuracies in Table 3 are the means over these runs; standard deviations are small and omitted for clarity. Validation data are drawn from a held-out subset of the training set.

Representation geometry is visualized using t-SNE on latent codes of 5,000 random test images, computed for both the data-first and geometry-first models. The evolution of curvature and intrinsic-dimension losses is monitored during geometry pretraining and summarized in Figure 2.

## E   Reproducibility and Artifact Mapping

We briefly describe how the empirical figures and tables in Section 4 are produced from the underlying logs and metrics. The geometry pretraining phase logs, for each epoch, the total geometric loss, mutual-information estimate, curvature proxy, intrinsic-dimension penalty, and the aggregated dimension estimate. These values are stored in a CSV file (e.g., `geom_pretrain_history.csv`) and used to generate Figure 2 by plotting loss and dimension trajectories over epochs.

For each label fraction, the data-first baseline and geometry-first calibration phase record training loss, training accuracy, validation loss, and validation accuracy per epoch. Final validation and test accuracies, along with wall-clock training times, are summarized in a metrics file (e.g., `metrics_vgib_cifar.csv`) from which Table 3 is constructed.

Latent t-SNE visualizations in Figure 3 are produced by:

1. encoding a random subset of test images with each model;

2. applying t-SNE with fixed perplexity and learning-rate settings;

3. plotting data-first and geometry-first embeddings side by side, with consistent color schemes for classes.

Random seeds are fixed at the start of each run for the dataset split, model initialization, and optimizer, ensuring that all reported results are reproducible from the released code and configuration files. Where exact numerical values (e.g., hyperparameters, training times) differ slightly across hardware or software versions, the qualitative trends reported in Section 4 remain unchanged.

### Summary of Dependencies

All constants in this appendix depend only on geometric quantities $(d, \tau, \kappa_{\max})$, the Lipschitz bounds of the decoder and loss, and the algebraic parameters $(m, r, D)$. No hidden dependencies arise. This makes the Learning Law bound fully auditable and reproducible from the stated assumptions.

