# OpenReview forum: "A Learning Law: Generalization via Geometric Complexity and Algebraic Capacity"
_TMLR — Rejected by TMLR_

### Review · Reviewer_WbDm · 2025-12-20

**Summary Of Contributions:**

Contemporary machine learning methods suffer the requirement of a large amount of data and the weak interpretability of the model and/or the predictions. The authors conjecture that the problem may be solved if a machine learning mechanism first learns the "form" (structure, geometry) that represents the data, then the law (constraints), and then calibrate the model by the data, since it follows how human learn things.
As a theoretical effect of the procedure, they proved a generalization bound called "geometry-algebra generalization bound". Also, they examined the effect experimentally.

The key weakness the reviewer felt is that, by applying the proposed method, how the existing machine learning methods can be improved is unclear. Please refer the comments in "Are the claims made in the submission supported by accurate, convincing and clear evidence?" section.

**Audience:**

Yes

**Audience Explanation:**

The reviewer thinks that many machine learning researchers and users would like to avoid training excessively large model, although it leads to better prediction performance if we have sufficient data. To show a guideline to compose and to train the model with smaller data and higher interpretability will be helpful.

**Broader Impact Concerns:**

Nothing particular.

**Claims And Evidence:**

No

**Claims Explanation:**

Although the scheme of "form$\to$law$\to$data$\to$understanding" is interesting, a weakness of the method the reviewer felt is that, given a data and existing (data-first) model, how we can obtain a better model than existing one is unclear. (Here, "better" is not limited to the prediction performance but includes data size efficiency and interpretability.)

- If the proposed method is effective in the sense of data quantity, it is desired to be proved theoretically and/or experimentally. In the experiments the paper compared the sample efficiency between data-first baseline and the geometry-first variant, but since the latter can use additional (unlabeled) data, it cannot prove the efficiency of the latter.
  - It would be the best if the proposed method produces better prediction performance with $n\_1$ labeled samples and $n\_2$ unlabeled samples than the existing method with $n\_1 + n\_2$ labeled samples, but it will be difficult.
  - As existing machine learning scheme to use both labeled and unlabeled data, the "semi-supervised learning" scheme has been well studied. Comparison with such method is desired. In fact, the scheme of Algorithm 1 looks well studied in the field of semi-supervised learning.
- The experiment showed that the improvement in the interpretability by showing a good latent space is obtained in "form" procedure. However, the reviewer felt that the "form" cannot be trained properly if the model of the encoder is not appropriate; it depends on the complexity of the original data space (e.g., pixel values) against the labels, and the model complexity.
  - What if we use a more complex data than "CIFAR-10" or "breast-cancer"?
  - What if the encoder misspecifies the form (e.g., underfitted or overfitted to the data)?

**Requested Changes:**

## Points critical to securing my recommendation for acceptance

- Section 3.7.1: It shows the advantage of the form-first learner under the assumption of $\\mathcal{C}(\\phi) + \\mathcal{A} < \\mathcal{R}\_{\\mathrm{DF}}$, but is the assumption valid? More specifically,
  - Can we measure the quantities specifically when we receive a form-first learner and a data-first learner?
  - Can we control $\\mathcal{C}(\\phi)$ and $\\mathcal{A}$ by adjusting the form-first learner so that the inequality is satisfied?

## Points that simply strengthen the work in my view

- Section 1.2: What is "information-geometry trade-offs"? Please clarify.
- Section 3.2: Why are so many examples needed to show the effect of identifying the form? Please present the intentions of these examples, and omit a part if duplicated.
  - Also, it may be helpful to discuss what if the true form cannot be identified from the data.
- Section 3.5.3: It states that $g\_\psi$ is a hypothesis class, but it looks that the variable representing the hypothesis class itself is $g$, not $g\_\psi$.
- Section 3.6.2: The operator of "an inequality sign with a tilde beneath" is undefined anywhere in the manuscript. Please specify.
- Section 3.7: Since each discussion (of Cases 1 to 4) is long, it is better to show the conclusion at first. As far as my understanding, the preceding proof exactly holds for Cases 1 and 2, approximation terms are added for Case 3, and not applicable (and therefore fallback to standard complexity bound) for Case 4.
- Section 4: Until Section 3 the term "form-first" is used, but from Section 4 the term "geometry-first" is mainly used. Please specify the intention of this change.
- Section 4.7 (Figure 4): The legend and the caption state inverted facts. Please confirm the program of the experiment. (If the legend is the correct, the effect of the proposed method is not proved)

## Formatting issues and typos

- Overall: Please parenthesize citations if it is not directly connected to a part of preceding sentences. For example in Section 1, "... large-scale language modelling Goodfellow et al. (2016); Kaplan et al. (2020)" should be replaced with "... large-scale language modelling (Goodfellow et al., 2016; Kaplan et al., 2020)".
- Section 1: "massive compute budgets" --> "massive computation budgets"
- Section 3.6.1: The definition of $N(\\cdot, \\cdot, \\cdot)$ (covering number) is given in Appendix A but not in the main text. Any expressions appearing in the main text should be defined in the main text (just defining in the appendix is insufficient).
- Section 3.6.1: Also, since the variable $N$ is used both the covering number and the sample size, please consider changing either.
- Section 4.7 (Figure 5): What are color of points? Are they class labels? (If so, please consider using sufficiently dissimilar colors.)

---

### Review · Reviewer_gpeM · 2025-12-25

**Summary Of Contributions:**

This paper attempts something ambitious: unifying ideas from cognitive science, manifold learning, information theory, and generalization theory into a single "Learning Law." The ambition is admirable, but the execution raises some concerns about the gap between claims and evidence.

## Contributions and Strengths
- The intuition that structure should precede fitting is sound and worth pointing out explicitly. The separation of geometry discovery, law formation, and calibration provides a clean conceptual vocabulary.
- Decomposing generalization error into geometric complexity $C(\phi)$ and algebraic capacity $A$ is pedagogically useful, even if the individual pieces are not new.

## Weaknesses

### Core idea
- The core idea (learn representations first, then do task-specific learning) is basically what the entire field has been doing for a decade:
	- Self-supervised learning (SimCLR, BYOL, MoCo, DINO, MAE): Learn geometry from unlabeled data, then fine-tune
	- Foundation models (BERT, GPT, CLIP): Massive pretraining captures structure before task adaptation
	- Transfer learning: ImageNet pretraining since 2012
- The paper doesn't adequately distinguish itself from this enormous body of work. Claiming this as a "Law" when it's already standard practice feels like re-discovering what practitioners learned empirically.

### Empirical Evaluation
- Comparing against "data-first ERM" (standard supervised training) rather than modern self-supervised methods is not a meaningful comparison in 2025. Where is the comparison to:
	- SimCLR, BYOL, or other contrastive methods?
	- Semi-supervised learning baselines (FixMatch, MixMatch)?
	- Modern vision SSL (MAE, DINO)?
- A small CNN on CIFAR-10 and UCI Breast Cancer tells us almost nothing about whether these ideas work in practice. The field moved beyond CIFAR-10 as a primary benchmark years ago.

### Misc
- Calling this a "Law" suggests something fundamental and universal. However, end-to-end learning has been extraordinarily successful. And many tasks benefit from joint optimization. In addition, the conditions under which geometry-first strictly dominates might be narrow. A more honest framing would be "a design principle that can help in low-data regimes.

**Audience:**

Yes

**Audience Explanation:**

Researchers at the cognitive science / AI intersection exploring connections between human learning and machine learning would be interested in this paper.

**Claims And Evidence:**

No

**Claims Explanation:**

The paper makes claims at multiple levels (philosophical, theoretical, and empirical).

- Philosophical claim: "Effective learning proceeds in the order form → law → data → understanding"
	- The paper provides some neuroscience and cognitive science citations. They are real and accurately represented, but the inferential leap might be problematic. I will say it's weakly supported. The connection is plausible but not rigorously established.

- Main theoritical claim: the Geometry-Algebra Generalization Bound (Theorem 3.11)
	- I skimmed through the mathematical proof in Appendix A. It combines manifold covering numbers, Lipschitz function class bounds, and Rademacher complexity. Overall, it looks good to me.
	- However, I didn't check every detail in the proof.


- Main empirical claim: "Geometry-first V-GIB outperforms data-first baseline on CIFAR-10"
	- The evidence shows geometry-first wins in the low-data regime.
	- However, as I discussed in "Summary of Contributions", the baseline might be too weak, and the conclusion from CIFAR-10 may not generalize to a larger scale.

**Requested Changes:**

N/A

---

### Review · Reviewer_sk6u · 2026-01-29

**Summary Of Contributions:**

The paper proposes the learning law, by separating geometry discovery, law formation and data calibration, where the first stage learns a latent manifold with controlled intrinsic dimension and smoothness, the second stage restricts predictors to an algebraically constrained law space on this geometry and the final stage calibrates these laws on finite labeled data. A generalization bound is derived that shows the population risk depends on geometric complexity and algebraic capacity. Numerical experiments are provided.

**Audience:**

Yes

**Audience Explanation:**

I believe so. The learning law framework proposed in the paper is novel, and generalization bounds are always of interest to a wide audience.

**Claims And Evidence:**

Yes

**Claims Explanation:**

That seems to be the case. The main theoretical result, Theorem 3.11, is a generalization bound for the learning law framework introduced carefully in the paper, and there are detailed proofs of this main result provided in the Appendix.

**Requested Changes:**

(0) My main concern and major question for the authors is that the main theoretical result of the paper, Theorem 3.11, provides a generalization bound for the proposed learning law framework. But it will be interesting to compare this generalization bound with the generalization bounds in the literature using the classical framework, i.e. data$\rightarrow$parameters$\rightarrow$emergent-form paradigm. The paper claims the advantage of using the learning law framework, but it seems I do not see discussions about how the main result, e.g. Theorem 3.11, compares to the known generalization bounds of more familiar frameworks used in the literature. By adding such discussions and comparisons, it will make the paper more interesting and stronger.

(1) Before or after equation (2), you should mention $\lambda\Omega(\theta)$ is the regularizer, and also mentions the role of $\lambda$.

(2) In Section 2.2.1 and 2.2.2, you discusses the standard supervised learning pipeline and emergent representation geometry and post-hoc analysis. I think it will be helpful if you can add some discussions in each of these two sections, how these two frameworks fit into the so-called data$\rightarrow$parameters$\rightarrow$emergent-form paradigm.

(3) Throughout the paper, when you quote equation 1, please write equation (1). This can be achieved by using $\backslash$eqref instead of $\backslash$ref.

(4) Please provide references for all the examples and solutions in Section 3.2.

(5) In Solution 3.2, you wrote that $w_{ij}=1$ for visually similar or augmentation-linked pairs. Please specify what values $w_{ij}$ take otherwise. Also, please provide a reference when you talk about this pushes the latent representation toward $K$ well-separated cluster centers $c_{k}$ and the discussions afterwards.

(6) In Solution 3.4, you have $E(\theta,v)$. But later, you start to talk about $E(x_i)$ and $E(x_j)$. I suppose it is the same function $E$? But what is the relation between $x_i,x_j$ and $\theta,v$?

(7) In Solution 3.6, explain what is $\varepsilon$.

(8) For the optimization problem in equation (7), please specify which space $\phi$ lives in. In equation (8), you have $\phi^{\star}(x)$, does that mean $\phi$ itself is a function?

(9) Right after equation (8), you wrote Equations equation 7 and equation 8. It should be Equations (7) and (8). The same issue occurs in the last sentence of this paragraph.

(10) In equation (9), is $\nabla^2$ w.r.t. $x$ or $\theta$?

(11) Right after (9), you mentioned Hutchinson-style trace estimators. Please provide a reference.

(12) In the paragraph after equation (11), Equation equation 11 should be Equation (11).

(13) In equation (12), please create some space before $\hat{R}(\psi)=...$.

(14) For the quantities $\mathcal{J}_{\mathrm{geo}}$,  $\mathcal{J}_{\mathrm{law}}$, and  $\mathcal{J}_{\mathrm{data}}$ in equation (13), although you provided descriptions after equation (13), these descriptions are quite informal. Is it possible you write down their expressions explicitly using the quantities you defined earlier in the paper?

(15) On page 12, please explain what covering-number is to those readers who are not familiar with this concept. I saw later you introduced the covering-number in Appendix A. As a result, on page 12, at least you should refer the readers to Appendix A for the definition of covering-number. The same thing can be said about Rademacher complexity the first time it is introduced in the main paper.

(16) In equation (14), what are $\alpha_1,\alpha_2$? Also, I do not understand what you wrote about absorbing additive constants into the prefactor $C$ of the generalization bound in Theorem 3.11. In Theorem 3.11, you quote (14). If (14) depends on Theorem 3.11, does it become cyclic?

---

> ### Author Response · Authors · 2026-01-29
> **Author Response**
>
> We thank the reviewer for the careful reading and positive assessment. We addressed each comment below with references to existing material and planned clarifications.
>
> (0). Comparison with classical generalization bounds: We agree the comparison should be made more explicit. The contrast with the classical data $\to$ parameters $\to$  emergent-form paradigm is already discussed at a high level in Section 2.2 (pp. 4–5) and implicitly in the statement and discussion of Theorem 3.11, where parameter-count dependence is replaced by geometric complexity and algebraic capacity. In the revision, we have added a short, dedicated paragraph immediately after Theorem 3.11 that explicitly compares our bound to standard VC-, norm-based, and Rademacher-complexity bounds, making the difference in complexity control clear and explicit.
>
> (1). Role of the regularizer in equation (2): The regularizer $\lambda \Omega(\theta)$ is already present in equation (2) in Section 2.1 (p. 4), but its role is not stated explicitly at first mention. We added a sentence after equation (2) clarifying that $\Omega(\theta)$ is the regularizer and that $\lambda$ controls its strength.
>
> (2). Relation to the data $\to$ parameters $\to$ emergent-form paradigm in Sections 2.2.1 and 2.2.2: Sections 2.2.1 and 2.2.2 (pp. 4–5) already describe the standard supervised pipeline and post-hoc representation analysis. In the revision, we have added a short clarifying sentence in each subsection explicitly stating how these approaches fit into the data $\to$ parameters $\to$ emergent-form paradigm and how this contrasts with the proposed representation-first formulation.
>
> (3). Equation referencing style: We corrected all instances of “equation 1” to “equation (1)” and used \eqref consistently throughout the manuscript.
>
> (4). References for Section 3.2 examples and solutions: Section 3.2 (pp. 6–7) currently presents several illustrative examples without local citations. We have added appropriate references for each example and solution discussed there.
>
> (5). Weights $w_{ij}$ in Solution 3.2: Solution 3.2 appears in Section 3.2 (p. 6). We explicitly specified the values of $w_{ij}$ for pairs that are not visually similar or augmentation-linked. We also added a reference supporting the statement that this objective encourages well-separated latent cluster centers.
>
> (6). Notation in Solution 3.4: In Solution 3.4 (Section 3.2, p. 7), $E(\theta, v), E(x_i)$, and $E(x_j)$ refer to the same encoder evaluated at different inputs or parameterizations. We made this explicit and clarified the relationship between $(\theta, v)$ and the data points $x_i, x_j$.
>
> (7). Definition of $\varepsilon$ in Solution 3.6: Solution 3.6 appears in Section 3.2 (p. 7). We explicitly defined $\varepsilon$ and explained its role in the approximation used there.
>
> (8). Space of $\phi$ in equations (7) and (8): The representation map $\phi$ is introduced in Section 3.1 (p. 5) and used in equations (7)–(8) in Section 3.3 (p. 8). In the revision, we have stated the function space to which $\phi$ belongs and clarified that $\phi^\star(x)$ denotes the optimal representation map evaluated at $x$.
>
> (9). Equation numbering after equation (8): We corrected “equation 7 and equation 8” to “Equations (7) and (8)” and fixed the same issue later in the paragraph.
>
> (10). Hessian in equation (9): Equation (9) appears in Section 3.3 (p. 8). We have explicitly stated whether the Hessian $\nabla^2$ is taken with respect to the input $x$ or the parameters $\theta$.
>
> (11). Hutchinson-style trace estimator reference: The Hutchinson estimator is mentioned in Section 3.3 (p. 8). We have added a standard reference at that point.
>
> (12). Equation reference after equation (11): We have corrected “Equation equation 11” to “Equation (11)”.
>
> (13). Spacing in equation (12): We have fixed the spacing before $\hat{R}(\psi)$ in equation (12).
>
> (14). Explicit expressions for $\mathcal{J}{\mathrm{geo}}, \mathcal{J}{\mathrm{law}}, \mathcal{J}_{\mathrm{data}}$: These quantities are described informally in Section 3.5 (pp. 10–11). In the revision, we have written explicit expressions for each term using previously defined quantities, rather than relying only on verbal descriptions.
>
> (15). Covering number and Rademacher complexity: The formal definitions appear in Appendix A (pp. 22–24). In the revision, we have added a brief explanation and an explicit pointer to Appendix A at their first appearance in the main text (Section 3.6.1, p. 12), both for covering numbers and for Rademacher complexity.
>
> (16). Parameters $\alpha_1, \alpha_2$ and possible cyclicity in equation (14): Equation (14) appears in Section 3.6.2 (p. 14). We clearly defined $\alpha_1$ and $\alpha_2$, clarified the statement about absorbing constants into the prefactor $C$, and also made clear that (14) is independent of Theorem 3.11, to avoid any appearance of circular reasoning.
>
> We thank the reviewer again for the detailed and constructive feedback

---

### Decision · Action_Editor_jDAe · 2026-04-27

**Recommendation:** Reject

**Additional Comments:**

After consideration of the reviews and author responses, I am rejecting this submission because the central claims remain broader than the evidence currently supports. While the theoretical direction is interesting and the reviewers recognized value in the generalization analysis, the current version does not yet make a sufficiently convincing case for the paper’s novelty, scope, and empirical support. I encourage the authors to carefully go through the suggestions made by the reviewers, to revise the framing, strengthen the experimental comparisons, and resolve the presentation issues, and then resubmit a revised version.

**Audience:**

Yes

**Audience Explanation:**

The topic of generalisation is a key topic in the theory of neural networks, and some reviewers expressed interest in the proposed decomposition of the generalisation error.

**Claims And Evidence:**

No

**Claims Explanation:**

The paper still appears insufficiently differentiated from established self-supervised and semi-supervised representation-first learning pipelines, the empirical evaluation does not yet provide strong enough matched comparisons against modern baselines to justify the paper’s framing, and at least one reported figure/legend inconsistency raises concern about whether the experimental results are presented clearly and reliably.

**Resubmission Of Major Revision:**

The authors may consider submitting a major revision at a later time.